# FORECASTBENCH: A DYNAMIC BENCHMARK OF AI FORECASTING CAPABILITIES

**Ezra Karger** *
Forecasting Research Institute
Federal Reserve Bank of Chicago
ezra@forecastingresearch.org

**Houtan Bastani** *
Forecasting Research Institute
houtan@forecastingresearch.org

**Chen Yueh-Han** *
New York University
yc7592@nyu.edu

**Zachary Jacobs**
Forecasting Research Institute
zach@forecastingresearch.org

**Danny Halawi**
University of California, Berkeley
dhalawi@berkeley.edu

**Fred Zhang**
University of California, Berkeley
z0@berkeley.edu

**Philip E. Tetlock**
Forecasting Research Institute
University of Pennsylvania
tetlock@wharton.upenn.edu

## ABSTRACT

Forecasts of future events are essential inputs into informed decision-making. Machine learning (ML) systems have the potential to deliver forecasts at scale, but there is no framework for evaluating the accuracy of ML systems on a standardized set of forecasting questions. To address this gap, we introduce **ForecastBench**: a dynamic benchmark that evaluates the accuracy of ML systems on an automatically generated and regularly updated set of 1,000 forecasting questions. To avoid any possibility of data leakage, ForecastBench is comprised solely of questions about future events that have no known answer at the time of submission. We quantify the capabilities of current ML systems by collecting forecasts from expert (human) forecasters, the general public, and LLMs on a random subset of questions from the benchmark ($N = 200$). While LLMs have achieved super-human performance on many benchmarks, they perform less well here: expert forecasters outperform the top-performing LLM ($p$-value $< 0.001$). We display system and human scores in a public leaderboard at www.forecastbench.org.

## 1 INTRODUCTION

Forecasting the future is a challenging but important task (Armstrong, 2001; Tetlock and Gardner, 2015). Economic forecasts influence investment and hiring decisions (Christensen et al., 2018). And forecasts of the Covid-19 pandemic in early 2020 prompted local lockdowns to slow the spread of the virus (Adam, 2020). However, human forecasting is often expensive, time-consuming, applicable only in specific domains, and subject to concerns about human biases. Motivated by these limitations, recent work explores the use of machine learning (ML) models, especially large language models (LLMs), in automated forecasting (Fluri et al., 2024; Halawi et al., 2024; Hendrycks et al., 2021; Phan et al., 2024; Pratt et al., 2024; Yan et al., 2024; Zou et al., 2022).

---

*Equal contribution.
Correspondence to forecastbench@forecastingresearch.org.
The views expressed in this paper do not necessarily reflect the views of the Federal Reserve Bank of Chicago or the Federal Reserve system.

To assess LLMs' forecasting capabilities, prior work built static benchmarks of questions where the answer was known (resolved) *after* the knowledge cut-offs of the LLMs they test (Halawi et al., 2024; Yan et al., 2024; Zou et al., 2022). For example, "Will a nuclear weapon be detonated in 2023," though resolved on Jan 1, 2024, is a valid out-of-sample question for testing a model with a known knowledge cut-off before the end of 2023.

This static evaluation methodology comes with three key drawbacks. First, as the knowledge cut-offs of frontier models are updated, static benchmarks become obsolete, necessitating further data-sourcing. This makes it difficult to continuously track and compare the top models in the field. Second, knowledge cut-offs are usually estimated using the time range of pre-training data. Such estimates may not be accurate, and post-training may inject further post-cutoff knowledge into the model, contaminating the test sets. Lastly, model developers face financial incentives to exaggerate their accuracy on benchmarks and claim that their models are state-of-the-art performers. While some fudging is easily identified, other subtle benchmark manipulation or overfitting is harder to catch, and some studies show significant evidence of benchmark contamination and/or memorization by popular LLM models (Elazar et al., 2024; Li et al., 2023; Roberts et al., 2023).

To address these drawbacks, we introduce **ForecastBench**, a dynamic benchmark that is continuously updated with new questions about future events. Our automated system gathers new questions from nine sources on a daily basis. These sources include prediction markets, forecasting platforms, and real-world time series. We regularly elicit predictions on these questions from both LLMs and human forecasters. As they resolve, we update a public leaderboard to show participant performance. This process makes ForecastBench an accurate real-time benchmark of forecasting ability.

Our initial benchmark consists of 1,000 standardized forecasting questions randomly sampled from a much larger real-time question bank. To establish baseline levels of performance, we record predictions from dozens of LLMs on the initial set, using methods like retrieval-augmentation to boost performance (Halawi et al., 2024; Lewis et al., 2020). We independently elicit predictions from two groups of human forecasters: the general public and seasoned forecasters (superforecasters) (Tetlock and Gardner, 2015) who have consistently performed well in competitive forecasting tournaments. As questions resolve, we rate the submissions of registered models and the human comparison groups, updating our public leaderboard.

Our initial results indicate that state-of-the-art models, such as Claude-3.5 Sonnet and GPT-4 Turbo, perform only roughly as well as a simple median of forecasts from a survey of humans with no (or minimal) forecasting experience, even when the LLMs are augmented with news retrieval and prompt engineering and when they have access to forecasts from a human crowd (on market-based questions). The models also perform significantly worse than the median forecast of superforecasters.

These findings leave significant room for researchers to improve AI-based forecasting systems using innovative approaches, such as developing methods for continuously updating models with current events and enhancing LLMs to reason over extended time frames. To support progress in this area, we publish an auxiliary dataset of LLM and human forecasts, rationales, and accuracy for use in future LLM fine-tuning and testing.

## 1.1 RELATED WORK

**Automated forecasting**  ML systems that make accurate forecasts can help inform human decision-making (Hendrycks et al., 2021; Schoenegger et al., 2024a). Recent work studies the use of LLMs for automated forecasting (Halawi et al., 2024; Jin et al., 2021; Pratt et al., 2024; Yan et al., 2024; Zou et al., 2022). These papers all build static benchmarks of resolved questions. A recent paper from Halawi et al. (2024) uses questions resolved between June 2023 and January 2024. Unfortunately, the latest LLMs have knowledge cut-offs past this point. This fact motivates our work to build a dynamically updating benchmark that can accurately evaluate frontier model performance.

In addition, Schoenegger and Park (2023) and Abolghasemi et al. (2023) compare the accuracy of GPT-4 and other LLMs to human forecasters. Schoenegger et al. (2024b) found that an ensemble of 12 LLMs rivaled human forecasts in a forecasting tournament with a small number of questions, limiting the study's statistical power. Unlike our work, that tournament was run only once and is no longer updated. Also, our much larger question set allows us to make precise statistical claims about the performance of LLMs relative to each other and to humans.

Finally, recent work focuses on the use of LLMs and transformer-based systems in statistical time-series forecasting (Das et al., 2024; Dooley et al., 2023; Goswami et al., 2024; Gruver et al., 2023; Jin et al., 2024; Nie et al., 2023; Rasul et al., 2023; Woo et al., 2024), but many of the most important forecasting questions do not have well-defined time series that can be used in standard statistical forecasting models (e.g., what is the probability that a nuclear weapon will be used offensively in the next ten years?). Our benchmark is more general, and evaluates forecasting performance across questions with and without underlying time series and historical baseline data.

**Language model evaluation**   Evaluating highly capable LLMs is a challenging task—with many models saturating key benchmarks soon after their release (Maslej et al., 2023; Owen, 2024) and with benchmarks potentially leaked to models' training data (Balloccu et al., 2024; Jacovi et al., 2023; Jiang et al., 2024b; Magar and Schwartz, 2022; Sainz et al., 2023; Xu et al., 2024a;b; Zhang et al., 2024). Our dynamic forecasting benchmark avoids both of these issues. First, automated forecasting is highly unsaturated: Halawi et al. (2024) showed that under simple zero-shot evaluation, frontier models such as GPT-4 are much less accurate than aggregates of human predictions. Second, our benchmark is dynamic. The final resolution of each question is only determined in the future and cannot be leaked in any training data (by construction). This eliminates concerns of contamination.

## 2  PRELIMINARIES

**Forecasting**   A forecasting question asks for a probabilistic prediction of a future event. A forecaster may assign probabilities to potential outcomes of the event. Forecasting platforms, including prediction markets, host a wide range of questions. Many questions are of public interest, such as "Will a Democrat win the 2028 US presidential election?"

**Metrics**   For binary questions, we use the Brier score as the performance metric, defined as $(f - o)^2$, where $f \in [0, 1]$ is the probabilistic forecast and $o \in \{0, 1\}$ is the outcome. Lower Brier scores are better, and a score of $0.25$ is associated with the uninformed forecast of $0.5$. Brier scores are strictly proper, incentivizing truthful reporting from participants.

**Models**   We evaluate 17 LLMs on our initial benchmark: GPT-3.5-Turbo-Instruct (Brown et al., 2020), GPT-4 (OpenAI, 2023), GPT-4o, Llama-2-70B (Touvron et al., 2023), Llama-3-7B, Llama-3-70B, Mistral-7B (Jiang et al., 2023), Mixtral-8x7B (Jiang et al., 2024a), Mixtral-8x22B, Mistral-Large, Qwen1.5-110B-Chat (Bai et al., 2023), Claude-2.1 (Anthropic, 2023), Claude-3-Haiku, Claude-3.5-Sonnet, Claude-3-Opus (Anthropic, 2024), Gemini 1.5 Flash, and Gemini 1.5 Pro (Gemini Team, 2023). We outline the various baselines we run with these models in Section 5.

## 3  BENCHMARK, LEADERBOARD, AND DATASETS

We maintain a question bank with a growing set of forecasting questions. Continuously adding questions to the question bank allows it to stay relevant and ensures that we have a large selection of unresolved questions to sample from.

Every night, our automated system updates the question bank with new questions and resolution values. We drop invalid, low-quality, and resolved questions, categorizing the remaining questions by topic. The process is fully automated, as detailed in Section 3.1.

Every two weeks, we sample from the question bank and release question sets for those interested in submitting their forecasts to the benchmark. We also survey a standard set of LLM-based models to allow for comparisons of performance over time. Submitted forecasts are resolved continuously with daily updates to our leaderboard. We provide the resulting data on forecast questions and submissions to researchers. See Section 3.2 for details. Finally, we discuss the question resolution procedure in Section 3.3 and our leaderboard design in Section 3.4.

### 3.1  QUESTION BANK

In an automated process that runs nightly at `0:00` UTC, questions are added to the question bank, resolution values are updated, and metadata generated. Details follow.

### 3.1.1 QUESTIONS AND RESOLUTION VALUES

We bring in questions from two broad types of sources: **markets** and **datasets**.[1] We select multiple, reliable sources from each type, ensuring the diversity of our benchmark. See Table 1 for details.

**Markets**    We compile market questions from a handful of prediction markets and forecast aggregation sites that elicit human predictions on questions across a wide range of topics.[2] When selecting questions from market sources to add to the question bank, we choose those with high levels of active human trading on these platforms (liquidity) as these questions tend to be of higher quality than those with low levels of human trading.[3] We store the latest market and resolution values for each question.

**Datasets**    We create dataset questions from well-established and well-maintained datasets that track real-world events (e.g., ACLED (Raleigh et al., 2023), a geopolitical database that tracks worldwide conflict, including facts like the number of protests in Niger each month). With these sources, we can generate questions using a fixed question template (e.g., "Will the number of protests in Niger increase by at least 10% over this month's value by the resolution date?").[4]

**Question Bank**    Table 1 lists the sources of market and dataset questions, along with the number of questions available in our question bank, whence we sample questions for our benchmark runs. In addition to the main questions (with sample size $N$), we also construct additional *combination* questions by choosing pairs of questions within each source. We describe combination questions in more detail in Section 3.2.

Table 1: Question bank composition, grouped by source type (market or dataset).

| Source | URL | $N$ | $\binom{N}{2}$ |
|---|---|---|---|
| RFI | randforecastinginitiative.org | 18 | 153 |
| Manifold Markets | manifold.markets | 405 | 81,810 |
| Metaculus | metaculus.com | 722 | 260,281 |
| Polymarket | polymarket.com | 915 | 418,155 |
| **Market Total** | | 2,060 | 760,399 |
| ACLED | acleddata.com | 3,220 | 5,182,590 |
| DBnomics | db.nomics.world | 52 | 1,326 |
| FRED | fred.stlouisfed.org | 166 | 13,695 |
| Wikipedia | wikipedia.org | 428 | 91,378 |
| Yahoo! Finance | finance.yahoo.com | 509 | 129,286 |
| **Dataset Total** | | 4,375 | 5,418,275 |
| **Question Bank Total** | | 6,435 | 6,178,674 |

### 3.1.2 QUESTION METADATA

After we automatically update the question bank with new forecasting questions and resolution values, the data are processed in several ways. This generates more information about the questions and creates another sampling option when creating the question set. Following Halawi et al. (2024), we use `gpt-4o-mini` to categorize questions by subject and to filter out low-quality questions. We report the number of questions by category and source in Table 15, where we display the breakdown of the standard questions from our question bank ($N$ from Table 1) across question categories by source, after removing low-quality questions.

---

[1]Licensing information can be found in Section C.1.

[2]Hereafter, for brevity, we refer to questions that come from both prediction markets and forecast aggregation sites as "market" questions. We also refer to theses sources collectively as "market" sources.

[3]See Table 8 for an example question pulled from a market source.

[4]See Table 9 for an example question pulled from a dataset source.

## 3.2 QUESTION SETS

**LLM question set**   We release a set of 1,000 forecast questions for LLMs every other Sunday at midnight UTC. We sample an equal number of questions from each source to ensure representativeness. Within each source, we then uniformly sample questions across all question categories, aiming for an equal distribution from each category within each source. This ensures that models cannot be overfit to a specific type of question or topic. Limiting the number of questions generated to 1,000 also ensures that costs for testing LLMs on the benchmark are capped for development teams with fewer resources.

**Human question set**   The human question set is comprised of 200 forecast questions sampled directly from the LLM question set. When sampling, we do our best to maintain proportionality across question sources and across categories within each question source; this ensures the question set addresses as many domains as possible.

**Forecast horizons**   For questions derived from dataset sources, the distribution of resolution dates is the $forecast\_due\_date + n$ days, where $n \in \{7, 30, 90, 180, 365, 1095, 1825, 3650\}$. In other words, for each dataset question we ask for 8 forecasts that differ only in their resolution date. For questions derived from market sources, we ask for only 1 forecast: the probability that each question will resolve positively (will the event underlying the question occur, or not). This setup will allow us to evaluate both human and LLM forecasting performance over the short, medium, and long term.

**Combination questions**   There are two types of questions, each comprising half of the question set. The first type is a standard forecasting question with a binary outcome, e.g., "Will inflation (core CPI) be above 3% next month?" We construct the second type, *combination* questions, by pairing two standard questions. For combination questions, we ask for forecasts on all Boolean combinations of the two questions (i.e., $P(Q1 \cap Q2)$, $P(Q1 \cap \neg Q2)$, ...). Considering the extensive number of standard questions and potential combinations in our question bank (as we could potentially combine 3, 4, or more standard questions together), we effectively have access to millions of possible forecasting questions from which we can sample. We show the number of two-question combination questions in the question bank as it stands at time of writing in the right-hand column of Table 1. This setup implies that for market combination questions, each forecaster will provide 4 forecasts, whereas for dataset combination questions, each forecaster will provide 32 forecasts (4 for each Boolean combination of $Q1$ and $Q2$ at each of the 8 forecast horizons).

Combination questions require forecasters to consider the covariance structure of different events, some of which are more independent than others. For instance, the best forecasts of whether the S&P500 (a key U.S. stock market index) will reach an all-time high and whether Spain will win the next Men's World Cup are likely independent. However, the best forecast of whether the S&P500 will reach an all-time high and whether the U.S. will enter an economic recession must account for the likely strong correlation between these events.

Of the 1,000 questions in the LLM question set, 500 are combination questions. Each combination question is composed of two standard questions from the same question set. This means that LLMs will also provide forecasts for the individual components of these combination questions, since they're drawn from the existing standard questions. Importantly, none of the 200 questions in the human question set are combination questions.[5]

**Timeline of forecasting round**   To compare human performance to LLMs, we periodically run surveys asking the general public and superforecasters to forecast on our question sets (see Section 4). We produce the question sets 10 days before the forecast due date to allow for time to create and run a human survey. LLM teams receive their question set 24 hours before the due date, even though it was generated at the same time as the human question set. This constrained time frame gives teams less time to be able to game the system. We thus elicit forecasts, obtaining comparable forecast sets on the due date from both LLMs and humans who faced the same information environment.

---

[5]We exclude combination questions because expert human forecasters are expensive; we maximize their relevance by having them focus on standard questions.

## 3.3 RESOLUTION

We resolve forecast sets nightly by gathering the latest information about which events have or have not occurred. All questions are ultimately resolved to ground truth.

**Evaluating performance on market questions**   For market questions, ground truth is not available until the question has been resolved on the platform. Until then, we evaluate performance by calculating the squared distance between the forecasted value and the platform's crowd forecast (an aggregate of human forecasts reported on the platform) from the preceding day.[6] This provides a good estimate of forecast performance on unresolved questions since crowd forecasts tend toward ground truth as the resolution date approaches.

Once a market question has officially been resolved, we score the forecast against the resolution value, creating a definitive score for the question. We are thus able to estimate forecast performance on the entire set of market questions (resolved and unresolved), incorporating all information available to markets on a nightly basis.

**Evaluating performance on dataset questions**   Datasets can be updated as new information becomes available. Hence, questions derived from datasets are continuously resolved to the value from the latest available data. As the resolution dates (ranging from a 7-day to a 10-year horizon) for dataset questions come due, a new round of forecasts is evaluated. We thus are able to evaluate forecasting performance over different time horizons.

**Missing forecasts**   We select 1,000 questions for the LLM question set to make the forecasting task impractical to complete manually within the 24-hour window after the question set is released. And we obligate all LLMs to forecast on all questions to ensure comparability of scores across models and human-based aggregates. When a model does not submit forecasts on certain questions or time horizons, we consider that a model error and impute a naive forecast for the model to ensure comparability across models over time.

For market questions, since we only ask for forecasts on the outcome of the question, missing forecasts are assigned the value of the crowd forecast on the forecast due date. Some may argue this is overly-generous, but we did not want forecasters to have a competitive advantage by simply scraping the market websites themselves.

For dataset questions, we impute the value $0.5$ (which represents an uninformed 50% forecast) to forecasts across all time horizons. Empirically, top models report valid forecasts on all questions and are not affected by this imputation procedure.

## 3.4 LEADERBOARDS

We generate leaderboards that rank models and humans by average overall score. The main leaderboard highlights the top forecasting submission across all questions and can by sorted by performance on the question type (market or dataset) and by resolution status (resolved or unresolved). The leaderboard is updated nightly after scoring forecasts against the latest data, market resolution values, and crowd forecasts.

## 3.5 DATASETS

As a key output of ForecastBench, we generate four datasets that grow over time.[7]

**General public forecast dataset**   Every time we run the public survey described in Section 4, we provide multiple independent forecasts and rationales for every one of the 200 forecast questions and report the accuracy of the median public forecast. See Section B.2.1 for details.

---

[6]The crowd forecasts are updated nightly in the Question Bank as described in Section 3.1.1.
[7]See Appendix A for licensing details and Appendix B for data dictionaries.

**Superforecaster forecast dataset**   In addition to forecasts and rationales, the superforecasters provide pertinent information about their forecasting process, like search terms used and useful URLs consulted. See Section B.2.2.

**LLM forecast dataset**   Similar to the general public dataset, we ask LLMs to produce forecasts on each of 1,000 forecast questions in the LLM question set. Their rationales are also included in the dataset whenever provided. See Section B.2.3 for details.

**Question & resolutions dataset**   In creating the benchmark, we have automated question creation and resolution from all of the sources outlined in Table 1. We provide these as a dataset that can be combined with the forecast datasets mentioned above. See Section B.1 for details.

## 4   HUMAN FORECASTER BASELINE

To compare LLM forecasting performance to human performance, we ran surveys of two different groups: the general public and superforecasters. We scored each group's median forecast, treating it as representative of its overall performance.

### 4.1   GENERAL PUBLIC

We recruited 500 human forecasters via Prolific and advertisements on Facebook to participate as representatives of the general public. These human subjects completed a brief introductory survey to gather demographic information[8] and evaluate performance on a few forecasting and comprehension tasks. They then completed a one-hour survey containing 20 random questions from the 200-item human question set described in Section 3.2, providing their forecasts and rationales for each question.

The number of responses per question varied to ensure representativeness across categories and sources; at least 40 responses were gathered per question, averaging 49 responses per question.

### 4.2   SUPERFORECASTERS

We recruited 39 superforecasters, who have a strong track record of accurate performance on a diverse set of geopolitical questions, to participate in a 9-day forecasting experiment in which participants were prompted to give their individual forecasts for 20 random questions from the same 200-item human question set described above. Roughly halfway through the 9-day experiment, participants were moved into a group forecasting stage in which we allowed them to see one another's forecasts and rationales and to communicate about each question. They were also given the opportunity to forecast on questions beyond the 20 questions assigned to them individually.

Because of the lower number of superforecasters, questions generally had fewer responses than in the public survey; a minimum of 3 forecasts were recorded for each question, with an average of 8 responses per question.[9]

## 5   LLM BASELINE

In this section, we evaluate the forecasting capabilities of LLMs and report on the methodology and results.

### 5.1   METHODOLOGY

We evaluate a suite of instruction-following chat models without any additional fine-tuning (see Section 2 for details on the models). For each baseline outlined below, we prompt the model to generate a probabilistic forecast that the question will resolve to "Yes."

---

[8]See Appendix L for an overview of public participant demographics.

[9]See Figure 2 for an example question from the human surveys.

**Baselines** We implement seven baselines: (1) zero-shot prompting; (2) prompting with scratchpad instructions; (3) prompting with scratchpad instructions and retrieved news articles; (4) zero-shot prompting with crowd forecasts; (5) scratchpad prompting with crowd forecasts; (6) scratchpad prompting with retrieved news articles and crowd forecasts; and (7) aggregating predictions from multiple LLMs. Each baseline is described in more detail below.

1 Our first baseline prompts the model **zero-shot** to generate a forecast directly without generating other content, such as intermediate thinking (Figure 4). By prompting the model to output its forecast directly, we assess raw forecasting capability without sensitivity to prompting strategies.

2 Our second baseline, prompts the model with **scratchpad** instructions (Nye et al., 2021) that outline a procedure the model should use to reason about the question (Figure 5). Our scratchpad prompt comes from (Halawi et al., 2024), which formed its prompts through a combination of analyzing the Brier score as prompt changes were made, and by adding language to fix common errors the LLMs would make, e.g., asking them to rephrase the question for understanding.

3 Since LLMs' knowledge is not continuously updated, it is important to provide them with up-to-date information relevant to the question (Zou et al., 2022). Our fourth baseline, **scratchpad with news**, uses the same scratchpad prompt as above, supplemented with retrieved news articles. The retrieval system is the same as described in Halawi et al. (2024): an LLM generates search queries for a news API, filters articles for relevancy, and summarizes the articles.

4 The question sets we provide to LLMs contain what we term **freeze values**. For market questions these are just the crowd forecast on the market the day the question set was created, as described in Section 3.2. For dataset questions, these are baseline values relevant to the forecasting task.[10] Our third baseline is the **zero-shot with freeze values**. This is simply the zero-shot prompt from Baseline 1 supplemented with the freeze value and an explanation of the freeze value. For examples of the freeze value and its explanation, see Table 8 and Table 9.

5 Our fifth baseline is the **scratchpad with freeze values** (the scratchpad prompt from Baseline 2 supplemented with freeze values as explained in Baseline 4).

6 Our sixth baseline is the **scratchpad with news with freeze values**.

7 In our final baseline, we aggregate the predictions generated by LLMs into an **LLM "ensemble"** forecast. We do this as Metaculus (2023) shows that an ensemble of all forecasters consistently outperforms using just the top 5, 10, ..., 30 best forecasters (based on past scores).

  To produce the LLM ensemble forecast, we use 3 models (GPT-4o, Claude-3.5-Sonnet, and Gemini-1.5-Pro) and 3 prompts crafted by superforecasters (Figure 6, Figure 7, and Figure 8). This results in 9 forecasts per question. We generate 3 LLM ensemble baselines using the median, geometric mean, and geometric mean of log odds (Satopää et al., 2014). For details, see Appendix E.

## 5.2 RESULTS

**Comparing humans and LLMs** In Table 2, we show that superforecasters achieve an overall mean Brier score of 0.096, significantly outperforming both the general public (Brier $= 0.121$, $p < 0.001$) and the top LLM performer on the 200-item subset (Claude 3.5 Sonnet: Brier $= 0.122$, $p < 0.001$).[11] The top-performing LLMs all had access to the crowd forecast on market questions (the "freeze values" from Baselines 4, 5, and 6 above). The top-performing model without access to the crowd forecast on market questions was less accurate than models with access to the human forecast with a Brier score of 0.136. The comparison between humans and LLMs relies on the 200 questions forecasted by humans, which is a random sub-sample of the 1,000 questions in the question set provided to LLMs (excluding combination questions).[12]

---

[10]For example, in a question generated from a Wikipedia page about whether a chess player's Elo rating will increase by a given date, the freeze value is the chess player's Elo rating on the question set generation date. An explanation of what the freeze value represents is also provided.

[11]See statistical note in Appendix G.

[12]Accuracy measures are based on more than 200 forecasts because human and LLM forecasters submitted multiple forecasts on each dataset question, one for each time horizon. The results presented here include forecasts over the 7-, 30-, 90-, and 180-day time horizons.

Table 2: LLM/Human Leaderboard (top 10)

| Model | Organization | Information provided | Prompt | Brier Score ↓ | | | Confidence Interval | Pairwise $p$-value comparing to No. 1 | Pct. more accurate than No. 1 |
|---|---|---|---|---|---|---|---|---|---|
| | | | | Dataset ($N$=422) | Market ($N$=76) | Overall ($N$=498) | | | |
| Superforecaster median forecast | ForecastBench | – | – | 0.118 | 0.074 | 0.096 | [0.076, 0.116] | – | 0% |
| Public median forecast | ForecastBench | – | – | 0.153 | 0.089 | 0.121 | [0.101, 0.141] | <0.001 | 22% |
| Claude-3-5-Sonnet-20240620 | Anthropic | Freeze values | Scratchpad | 0.138 | 0.107 | 0.122 | [0.099, 0.146] | <0.001 | 31% |
| Claude-3-5-Sonnet-20240620 | Anthropic | News with freeze values | Scratchpad | 0.142 | 0.112 | 0.127 | [0.104, 0.150] | <0.001 | 29% |
| GPT-4-Turbo-2024-04-09 | OpenAI | Freeze values | Zero shot | 0.162 | 0.095 | 0.128 | [0.105, 0.151] | <0.001 | 32% |
| Claude-3-5-Sonnet-20240620 | Anthropic | Freeze values | Zero shot | 0.145 | 0.117 | 0.131 | [0.103, 0.159] | <0.001 | 31% |
| GPT-4 | OpenAI | Freeze values | Zero shot | 0.167 | 0.096 | 0.132 | [0.109, 0.155] | <0.001 | 31% |
| GPT-4o | OpenAI | News with freeze values | Scratchpad | 0.162 | 0.105 | 0.133 | [0.113, 0.154] | <0.001 | 25% |
| Claude-3-5-Sonnet-20240620 | Anthropic | – | Scratchpad | 0.138 | 0.133 | 0.136 | [0.113, 0.158] | <0.001 | 28% |
| GPT-4o | OpenAI | Freeze values | Scratchpad | 0.161 | 0.113 | 0.137 | [0.115, 0.158] | <0.001 | 27% |

*Notes:*

1. Shows performance on the 200 standard questions provided in the human question set at the 7-, 30-, 90-, and 180-day forecast horizons. See Table 18 for top 50.
2. The full leaderboard is available at www.forecastbench.org. Online results are updated nightly, so may be slightly different than the version presented here.
3. For resolved market questions, forecasts are compared against ground truth while for unresolved market questions, they are compared to community aggregates.
4. The overall score is calculated as the average of the mean dataset Brier score and the mean market Brier score.
5. Pairwise $p$-value comparing to No. 1 (bootstrapped): The $p$-value calculated by bootstrapping the differences in overall score between each model and the best forecaster under the null hypothesis that there's no difference.
6. Pct. more accurate than No. 1: The percent of questions where this forecaster had a better overall score than the best forecaster.

As a particular failure mode, we find LLMs are significantly worse at combination questions. Although our human surveys did not explicitly ask for forecasts on combination questions, we bound human performance by assuming independence of the component of each combination question. This underestimates human accuracy because a human forecaster predicting the outcome of a combination question could account for dependence between the permuted events. In Table 20, we present this comparison of human and LLM forecasts. We see that LLMs perform poorly on these combination questions, and including them in the benchmark widens the gap between human and LLM performance: superforecasters (Brier = 0.076) outperform the general public (Brier = 0.096) and the top LLM (GPT-4o, Brier = 0.130) significantly. To benchmark the size of this gap in performance, the 0.054 Brier score gap in performance between superforecasters and GPT-4o is significantly larger than the 0.026 gap in performance between GPT-4o and GPT-4.

Table 3: LLM Leaderboard (top 10)

| Model | Organization | Information provided | Prompt | Brier Score ↓ | | | Confidence Interval | Pairwise $p$-value comparing to No. 1 | Pct. more accurate than No. 1 |
|---|---|---|---|---|---|---|---|---|---|
| | | | | Dataset ($N$=5,492) | Market ($N$=897) | Overall ($N$=6,389) | | | |
| Claude-3-5-Sonnet-20240620 | Anthropic | Freeze values | Scratchpad | 0.169 | 0.078 | 0.123 | [0.117, 0.129] | – | 0% |
| GPT-4-Turbo-2024-04-09 | OpenAI | Freeze values | Scratchpad | 0.172 | 0.080 | 0.126 | [0.120, 0.132] | 0.096 | 43% |
| GPT-4o | OpenAI | Freeze values | Scratchpad | 0.186 | 0.069 | 0.128 | [0.122, 0.133] | <0.01 | 43% |
| Gemini-1.5-Pro | Google | Freeze values | Scratchpad | 0.162 | 0.106 | 0.134 | [0.128, 0.139] | <0.001 | 35% |
| GPT-4o | OpenAI | News with freeze values | Scratchpad | 0.190 | 0.084 | 0.137 | [0.131, 0.143] | <0.001 | 39% |
| Gemini-1.5-Pro | Google | News with freeze values | Scratchpad | 0.166 | 0.111 | 0.139 | [0.133, 0.144] | <0.001 | 34% |
| Claude-3-Opus-20240229 | Anthropic | Freeze values | Zero shot | 0.186 | 0.093 | 0.139 | [0.133, 0.146] | <0.001 | 41% |
| Qwen1.5-110B-Chat | Qwen | Freeze values | Scratchpad | 0.176 | 0.108 | 0.142 | [0.136, 0.148] | <0.001 | 30% |
| Claude-3-5-Sonnet-20240620 | Anthropic | News with freeze values | Scratchpad | 0.184 | 0.101 | 0.143 | [0.137, 0.149] | <0.001 | 32% |
| Claude-3-5-Sonnet-20240620 | Anthropic | Freeze values | Zero shot | 0.192 | 0.094 | 0.143 | [0.136, 0.150] | <0.001 | 42% |

*Notes:*

1. Shows performance on the 1,000 (500 standard, 500 combination) questions in the LLM question set at the 7-, 30-, 90-, and 180-day forecast horizons. See Table 19 for top 50.
2. The full leaderboard is available at www.forecastbench.org. Online results are updated nightly, so may be slightly different than the version presented here.
3. For resolved market questions, forecasts are compared against ground truth while for unresolved market questions, they are compared to community aggregates.
4. The overall score is calculated as the average of the mean dataset Brier score and the mean market Brier score.
5. Pairwise $p$-value comparing to No. 1 (bootstrapped): The $p$-value calculated by bootstrapping the differences in overall score between each model and the best forecaster under the null hypothesis that there's no difference.
6. Pct. more accurate than No. 1: The percent of questions where this forecaster had a better overall score than the best forecaster.

**Comparing LLMs** Table 3 excludes humans and evaluates LLMs on the entire question set ($N$=1,000 questions). Here we see a similar ranking of models, with Claude 3.5 Sonnet slightly outperforming GPT-4 Turbo. As in Table 2, most of the top-performing models use the scratchpad prompt (Figure 5) and use as inputs the human crowd forecasts for market questions. Access to recent topical news related to the questions did not improve performance.

**LLM performance and forecasting accuracy** Figure 1a demonstrates the seemingly linear relationship between Chatbot Arena scores (Chiang et al., 2024) and the overall Brier score from Table 2. We observe a significant correlation ($r = -0.68$, $p = 0.003$), indicating that models with higher Arena scores tend to produce more accurate forecasts. The linear relationship implies that LLMs

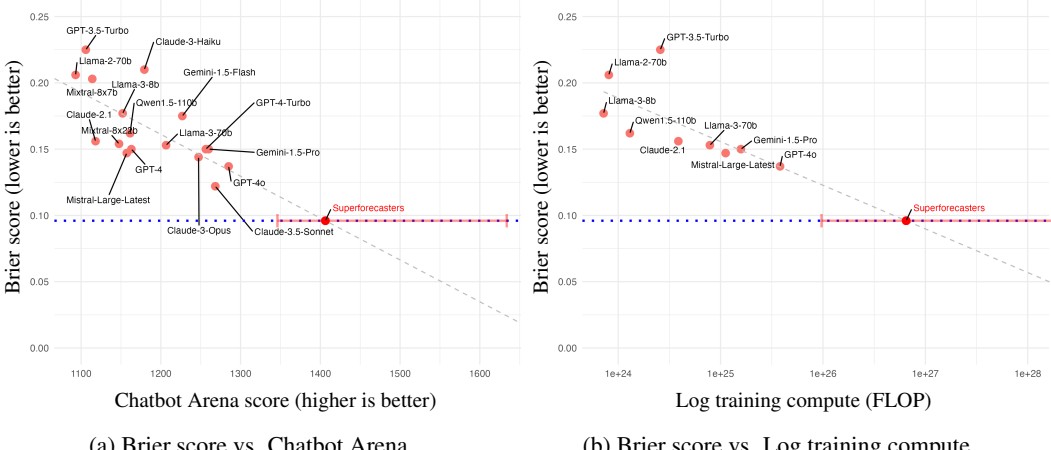

(a) Brier score vs. Chatbot Arena        (b) Brier score vs. Log training compute

Figure 1: The graphs show the linear relationship between the Brier scores from Table 2 and **(a)** Chatbot Arena scores and **(b)** estimates of training compute. The dotted blue line represents the Superforecasters' overall Brier score. A red dot with a bootstrapped 95% confidence interval is placed at the intersection of this dotted blue line with the dashed linear fit line to demonstrate the potential intersection of LLM Arena score/training compute and Superforecaster-level forecasting performance. For **(b)**, if estimates from Epoch AI (2024) were not available, we produced estimates following https://epoch.ai/blog/estimating-training-compute. The trend-line in **(a)** is $y = 0.506 - 0.000298x$ ($R^2 = 0.47$) and in **(b)** it is $y = 0.844 - 0.01213x$ ($R^2 = 0.41$).

could match superforecaster performance when the Arena score approaches $1406$ (bootstrapped 95% CI: 1346–1633).

Figure 1b shows the log-linear relationship between estimated training compute and the overall Brier score from Table 2. Projecting out the log-linear relationship, we find that LLMs could match superforecaster performance when training compute approaches $6.49 \times 10^{26}$, though there is a large confidence interval (bootstrapped 95% CI: $9.69 \times 10^{25}$–$8.65 \times 10^{28}$) given the marginally significant relationship ($r = -0.67$, $p = 0.046$).

## 6    DISCUSSION

We introduced ForecastBench, a dynamic and continuously updated benchmark for evaluating LLM forecasting capabilities. By focusing exclusively on questions that are unresolved at the time of submission, we eliminate the risks of data leakage and ensure a robust evaluation environment. Our initial results demonstrate that while state-of-the-art LLMs exhibit promising potential, they underperform superforecasters. This performance gap highlights the challenges in leveraging current LLMs for accurate, real-time forecasting.

We produce a public leaderboard listing the real-time accuracy of top LLMs and humans as well as a standardized dataset of forecasting questions and rationales. Future work should leverage this auxiliary dataset of predictions and rationales to fine-tune models, explore new architectures, and develop adaptive systems better suited for general reasoning in dynamic, real-world environments. Ultimately, ForecastBench serves as a step toward harnessing the full potential of AI-based systems for forecasting and decision-making.

## 7    REPRODUCIBILITY STATEMENT

One reason we've open-sourced our code (link in Appendix A) is to allow for independent verification of our results. See Appendix I for reproducing the human forecast sets, Appendix J for reproducing LLM forecast sets, and Appendix K for resolving the forecasts and creating the leaderboard.

## 8 ETHICS STATEMENT

Human survey subjects in both the public and superforecaster surveys are made aware prior to their participation in the study via an informed consent form (approved by our IRB, number 855431) that their forecast/rationale data may be publicly released and used to train large language models or other AI systems, with said data carefully reviewed and anonymized.

We have manually reviewed text provided by human participants to ensure that no personally identifiable information is released as part of our human forecast datasets, per IRB requirements. Similar manual reviews of text data will take place as part of every future human forecasting round.

## ACKNOWLEDGMENTS

We thank Open Philanthropy for providing a grant to fund this work, allowing us to maintain the benchmark at least until mid-2027.

We thank Otto Kuusela for helpful discussions, Sam Glover and Molly Hickman for writing super-forecaster prompts, and Harrison Durland and Victoria Schmidt for their research assistance.

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

## A    LICENSING AND MAINTENANCE

**Hosting**    The latest leaderboards, updated nightly, are available on www.forecastbench.org. Documentation is available on the repository wiki: github.com/forecastingresearch/forecastbench/wiki.

**Datasets**    Our datasets, distributed under the CC BY-SA 4.0 license, are available on www.forecastbench.org/datasets.html. Historical updates to the resolution datasets and leaderboards are available on github.com/forecastingresearch/forecastbench-datasets. Bi-weekly question sets are also released via this repository. The repository is mirrored to Hugging Face and available at huggingface.co/datasets/forecastingresearch/forecastbench-datasets.

**Codebase**    The code underlying our automated system runs on Google Cloud Platform and is available at github.com/forecastingresearch/forecastbench under the MIT license.

**Participating**    Bi-weekly forecasting rounds are open to LLM teams. Instructions for participating are can be found on the wiki.

**Maintenance & long-term preservation**    We ensure the long-term availability and maintenance of the benchmark as it is funded by Open Philanthropy until mid-2027. If no further funding is provided beyond that point, datasets will continue to be made available on GitHub and Hugging Face.

## B    DATASETS

We intend our datasets to be used for training general LLMs, fine-tuning forecasting LLMs, and for any applicable research purposes. No restrictions are placed on who may use our datasets, nor to what end.

**Availability**    www.forecastbench.org/datasets.html

### B.1    QUESTION AND RESOLUTION SETS

Every question set will be published. Their resolutions will also be published such that there's a complete training set when combined with the forecast sets outlined in Section B.2. The data dictionary for the question set is outlined in Table 4 and Table 5. The data dictionary for the resolution set is outlined in Table 6 and Table 7.

**Data format**    The question and resolution datasets are released as JSON (`.json`) files.

**Ethical and responsible use**    There are no restrictions on use of the question and resolution datasets.

**Data collection**    Our question and resolution datasets have been pulled, and are updated, from various, public-facing sources. From those sources where the terms of use/service prohibit the redistribution of their information (currently, Manifold Markets and Metaculus), we have obtained explicit permission to do so. Before we add new sources to our growing dataset, we will ensure the ability to distribute questions and resolutions publicly. Data sources in our question bank can be found in Table 14.

#### B.1.1    EXAMPLES

Table 8, Table 9, and Table 10 show concrete examples of the data dictionary detailed in Table 4.

### B.2    FORECAST SETS

Every set of forecasts provided to ForecastBench is made public and all forecasts coming from the general public and superforecasters are anonymized before release.

Each forecast set contains the header information outlined in Table 11, with all forecasts in an array called `forecasts`. The forecast sets are described in the following subsections.

Table 4: Question set data dictionary.

| Field | Description | Required | Data Type |
|---|---|---|---|
| `forecast_due _date` | Date in ISO format. e.g. `"2024-07-21"` | ✓ | string |
| `question_set` | The name of the file that contains the question set. e.g. `"2024-07-21-llm.json"` | ✓ | string |
| `questions` | A list of questions to forecast on, as defined in Table 5. | ✓ | array<object> |

**Data format**    The question and resolution datasets are released as JSON (`.json`) files.

**Ethical and responsible use**    There are no restrictions on use of the forecast sets.

### B.2.1    GENERAL PUBLIC FORECAST SET

This forecast set consists of both forecasts made by individuals on the given question set and of the aggregation of those forecasts, as described in Appendix D. The dataset will be updated every time a survey is run with every forecast containing the information outlined in Table 11 and Table 12. Note that the aggregated forecast sets do not contain the `user_id` or `reasoning` fields.

**Data collection**    Data is collected from human forecasters via Qualtrics. We collect the rationale behind forecasts and manually anonymize the data to ensure there is no personally identifiable information in the dataset before posting it online.

### B.2.2    SUPERFORECASTER FORECAST SET

In addition to the fields outlined in Table 11 and Table 12, the superforecaster dataset contains forecasts with the fields shown in Table 13 (minus the `suspected_LLM` field). Likewise, this dataset will be updated every time a group of superforecasters is surveyed.

**Data collection**    39 superforecasters provided forecasts, rationales, and additional information about they way they forecast in our latest survey round. We will further manually check all of their responses to ensure the anonymity of the dataset.

### B.2.3    LLM FORECAST SET

This dataset provides the same data (aside from `user_id`) as outlined in Table 11 and Table 12, only provided by language models. Each individual `.json` file was created by a model for the given question set.

**Data collection**    Beyond informing teams their forecasts will be made public, we will not check the rationales.

Table 5: Data dictionary for question entries in questions array from Table 4.

| Field | Description | Required | Data Type |
|---|---|---|---|
| id | A unique identifier string given source. If instead of a string it's an array of strings, then this is a combination question and combination_of will contain one question per id in the array of strings. | ✓ | string \| array<string> |
| source | Where the data comes from. | ✓ | string |
| question | The question to forecast, presented as an f-string with placeholders {forecast_due_date} and {resolution_date} for dataset questions. | ✓ | string |
| resolution_ criteria | ForecastBench resolution criteria. Specifies how forecasts will be evaluated for each question type. | ✓ | string |
| background | Background information about the forecast question provided by the source, if available. Default: 'N/A' | ✓ | string |
| market_info_ open_ datetime | The datetime when the forecast question went on the market specified by source. Default: 'N/A' | ✓ | string |
| market_info _close_ datetime | The datetime when the forecast question closes on the market specified by source. Default: 'N/A' | ✓ | string |
| market_info_ resolution_ criteria | The resolution criteria provided by the source, if available. Default: 'N/A' | ✓ | string |
| url | The URL where the resolution value is found. | ✓ | string |
| freeze_datetime | The datetime UTC when this question set was generated. | ✓ | string |
| freeze_datetime_ value | The latest value of the market or comparison value the day the question set was generated. | ✓ | string |
| freeze_datetime_ value _explanation | Explanation of what the value specified in freeze_datetime_value represents. | ✓ | string |
| source_intro | A prompt that presents the source of this question. | ✓ | string |
| combination_of | An array of question objects, as defined by this data dictionary. Default: 'N/A' | | string \| array<object> |
| resolution_dates | The resolution dates for which forecasts should be provided for this forecast question. | ✓ | array<string> |

Table 6: Resolution set data dictionary.

| Field | Description | Required | Data Type |
|---|---|---|---|
| `forecast_due _date` | Date in ISO format. e.g. `"2024-07-21"` | ✓ | string |
| `question_set` | The name of the file that contains the question set. e.g. `"2024-07-21-llm.json"` | ✓ | string |
| `resolutions` | A list of resolutions to the forecast questions, as defined in Table 7. | ✓ | array<object> |

Table 7: Data dictionary for resolution entries in questions array from Table 6.

| Field | Description | Required | Data Type |
|---|---|---|---|
| `id` | A unique identifier string given `source`. If instead of a string it's an array of strings, then this is a combination question. | ✓ | string \| array<string> |
| `source` | Where the data comes from. | ✓ | string |
| `direction` | If `id` has an array value, this is an array of the same length. Each entry $\in \{-1, 1\}$. If the value is 1, the question was asked in the normal direction. If the value is $-1$, the question was negated in the combination question e.g., for a question asking for $P(\neg Q1 \cap Q2)$, the value would be $[-1, 1]$; all possible directions for this question would be: $[1, 1], [-1, 1], [1, -1], [-1, -1]$. The value is `null` when `id` is a string. | ✓ | string |
| `forecast_due _date` | The date the forecast is due in ISO 8601 format `YYYY-MM-DD`. | ✓ | string |
| `resolution_date` | The date the value is associated with in ISO 8601 format `YYYY-MM-DD`. | ✓ | string |
| `resolved_to` | The resolution value for the given date. | ✓ | number |
| `resolved` | If `true` the question has been resolved. `False` otherwise. | ✓ | boolean |

Table 8: Prediction market example.

| Field | Entry |
|---|---|
| id | 1558 |
| source | metaculus |
| combination_of | N/A |
| question | Will the Cavendish account for less than 50% of banana exports worldwide before 2035? |
| background | Bananas are a well-liked import fruit all over the world, and the Cavendish cultivar has been crushing that market for sixty years. But its rise is literally founded upon the compost heap of the Gros Michel, another cultivar. The so-called 'Big Mike' variety had been the leading export towards Europe and North America, but the Panama disease, a fungus belonging to the \*Fusarium\* clade, killed that. Luckily the Cavendish, grown in the same soil as the wilting Gros Michel, replaced it as \*the\* banana most of the western world connected with bananas. However, it appears another \*Fusarium\* rears its spores. Cavendish, with their genetic homogeneity (they're all clones) and sterile nature, aren't resistant to it, and the fungus is ravaging more and more plantations. There are efforts under way to deal with \*Fusarium\*, but with various societies' doubts and misgivings about GMOs, the cure may be viewed as a curse instead. |
| market_info_ resolution_ criteria | This question will resolve as \*\*Yes\*\* if the Cavendish banana accounts for less than 50% of worldwide annual banana exports in any year from 2018 to 2034 (inclusive). |
| market_info_ open_datetime | 2018-11-13T08:00:00+00:00 |
| market_info_ close_datetime | 2034-12-31T23:00:00+00:00 |
| resolution_ criteria | Resolves to the outcome of the question found at https://www.metaculus.com/questions/1558/cavendish-bananas-collapse-by-2035/. |
| url | https://www.metaculus.com/questions/1558/cavendish-bananas-collapse-by-2035/ |
| freeze_datetime_ value | 0.48 |
| freeze_datetime_ value_explanation | The community prediction. |
| freeze_datetime | 2024-07-12T00:00:00+00:00 |
| source_intro | We would like you to predict the outcome of a prediction market. A prediction market, in this context, is the aggregate of predictions submitted by users on the website Manifold. You're going to predict the probability that the market will resolve as 'Yes'. |
| resolution_dates | N/A |

Table 9: Data source (DBnomics) example.

| Field | Entry |
|---|---|
| id | meteofrance_TEMPERATURE_celsius.07005.D |
| source | dbnomics |
| combination_of | N/A |
| question | What is the probability that the daily average temperature at the French weather station at Abbeville will be higher on resolution_date than on forecast_due_date? |
| background | The history of Average temperature by day and by station for France - Degree Celsius - ABBEVILLE - Daily from Météo-France is available at https://db.nomics.world/meteofrance_TEMPERATURE_celsius.07005.D. |
| market_info_ resolution_ criteria | N/A |
| market_info_ open_datetime | N/A |
| market_info_ close_datetime | N/A |
| url | https://db.nomics.world/meteofrance_TEMPERATURE_celsius.07005.D |
| resolution_ criteria | Resolves to the value found at https://db.nomics.world/meteofrance_TEMPERATURE_celsius.07005.D once the data is published. |
| freeze_datetime_ value | 17.95 |
| freeze_datetime_ value_explanation | The daily average temperature at the French weather station at Abbeville. |
| freeze_datetime | 2024-07-12T00:00:00+00:00 |
| source_intro | DBnomics collects data on topics such as population and living conditions, environment and energy, agriculture, finance, trade and others from publicly available resources, for example national and international statistical institutions, researchers and private companies. You're going to predict how questions based on this data will resolve. |
| resolution_dates | ["2024-07-28", "2024-08-20", "2024-10-19", "2025-01-17", "2025-07-21", "2027-07-21", "2029-07-20", "2034-07-19"] |

Table 10: Combination (DBnomics) example.

| Field | Entry |
|---|---|
| id | ["meteofrance_TEMPERATURE_celsius.07117.D", "meteofrance_TEMPERATURE_celsius.07240.D"] |
| source | dbnomics |
| combination_of | [An array containing dictionary entries of both questions.] |
| question | We are presenting you with two probability questions. Please predict the probability that both will happen, that one will happen but not the other, and that neither will happen. In other words, for each resolution date please provide 4 predictions. |
| background | N/A |
| market_info_resolution_criteria | N/A |
| market_info_open_datetime | N/A |
| market_info_close_datetime | N/A |
| url | N/A |
| resolution_criteria | N/A |
| freeze_datetime value | N/A |
| freeze_datetime value_explanation | N/A |
| freeze_datetime | 2024-07-12T00:00:00+00:00 |
| human_prompt | We are presenting you with two probability questions. Please predict the probability that both will happen, that one will happen but not the other, and that neither will happen. In other words, for each resolution date please provide 4 predictions. |
| resolution_dates | ["2024-07-28", "2024-08-20", "2024-10-19", "2025-01-17", "2025-07-21", "2027-07-21", "2029-07-20", "2034-07-19"] |

Table 11: Data dictionary of headers for forecast set.

| Field | Description | Required | Data Type |
|---|---|---|---|
| organization | The organization name as it should be displayed on the leaderboard. | ✓ | string |
| model | The model name as it should be displayed on the leader board. | ✓ | string |
| question_set | The name of the question set file these forecasts are associated with. | ✓ | string |
| forecast_due_date | The date the forecasts were due in ISO 8601 format YYYY-MM-DD. | ✓ | string |
| forecasts | All forecasts for this question set. | ✓ | array<object> |

Table 12: Public forecast set data dictionary of entries in `forecasts` array from Table 11.

| Field | Description | Required | Data Type |
|---|---|---|---|
| `id` | A unique identifier string given `source`. If instead of a string it's an array of strings, then this is a combination question. | ✓ | string \| array\<string\> |
| `source` | Where the data comes from. | ✓ | string |
| `forecast` | The forecast $\in [0, 1]$. | ✓ | number |
| `resolution_date` | The resolution date this forecast corresponds to. `null` for market questions. | ✓ | string \|`null` |
| `reasoning` | The rationale underlying the forecast. During data anonymization, we insert `[redacted to maintain anonymity]` wherever text has been redacted. | ✓ | string |
| `direction` | If `id` has an array value, this is an array of the same length. Each entry is an integer $\in \{-1, 1\}$. If the value is 1, the question was asked in the normal direction. If the value is $-1$, the question was negated in the combination question e.g., for a question asking for $P(\neg Q1 \cap Q2)$, the value would be $[-1, 1]$. All possible values are: $[1, 1]$, $[-1, 1]$, $[1, -1]$, $[-1, -1]$, and `null`. | ✓ | array\<number\> \|`null` |
| `user_id` | A randomly generated string associated with the human respondent who submitted the forecast. This value contains no information which could identify said participant and was assigned to the dataset after personal identifiers had been removed. Only required for participants from the general public and superforecaster surveys. | | string |

Table 13: Superforecaster forecast set data dictionary of entries in `forecasts` array from Table 11 (additional fields to those in Table 12).

| Field | Description | Required | Data Type |
|---|---|---|---|
| `searches` | An array of search terms used in researching the topic. | ✓ | array\<string\> \|`null` |
| `consulted_urls` | A list of useful URLs | ✓ | array\<string\> \|`null` |

## C   QUESTION BANK

### C.1   LICENSES

The licenses outlined in Table 14 apply to the datasets we have sourced for questions. We were granted express permission to use questions from Manifold Markets and Metaculus. Though not required by their license, we also met with a representative from ACLED who approved our use of their dataset for the benchmark and dataset distribution.

Table 14: Question sources and permissions

|  | License | License grants permission to use | Express permission granted |
|---|---|---|---|
| RAND Forecasting Initiative | Public Domain | ✓ |  |
| Manifold | Terms of Service | ✗ | ✓ |
| Metaculus | Terms of Use | ✗ | ✓ |
| Polymarket | Terms of Service | ✓ |  |
| ACLED | Terms of Use | ✓ | ✓ |
| DBnomics | Open License | ✓ |  |
| FRED | Terms of Use | ✓ |  |
| Wikipedia | Terms of Use | ✓ |  |
| Yahoo! | Terms of Use | ✓ |  |

### C.2   CATEGORIES

We generate metadata on all of the questions in our question bank, categorizing our questions, as described in Section 3.1.2.

It is important to have forecasting questions across a broad array of categories to test LLM capabilities. We are currently adding more questions from datasets with the goal of equilibrating these categories.

Table 15: Categories and question counts by source, dropping invalid questions.

|  | RFI | Manifold | Metaculus | Polymarket | ACLED | DBnomics | FRED | Wikipedia | Yahoo! | Total |
|---|---|---|---|---|---|---|---|---|---|---|
| Arts & Recreation | 0 | 42 | 10 | 65 | 0 | 0 | 0 | 0 | 0 | 117 |
| Economics & Business | 2 | 13 | 55 | 154 | 0 | 0 | 166 | 0 | 509 | 899 |
| Environment & Energy | 0 | 2 | 37 | 7 | 0 | 52 | 0 | 0 | 0 | 98 |
| Healthcare & Biology | 0 | 8 | 71 | 3 | 0 | 0 | 0 | 215 | 0 | 297 |
| Politics & Governance | 3 | 16 | 128 | 188 | 0 | 0 | 0 | 0 | 0 | 335 |
| Science & Tech | 5 | 66 | 172 | 15 | 0 | 0 | 0 | 1 | 0 | 259 |
| Security & Defense | 3 | 9 | 109 | 12 | 3,220 | 0 | 0 | 0 | 0 | 3,353 |
| Sports | 0 | 65 | 18 | 468 | 0 | 0 | 0 | 137 | 0 | 688 |
| Other | 6 | 151 | 122 | 2 | 0 | 0 | 0 | 75 | 0 | 356 |
| Total | 19 | 372 | 722 | 914 | 3,220 | 52 | 166 | 428 | 509 | 6,402 |

## D    HUMAN SURVEY INSTRUCTIONS AND SCREENSHOTS

### D.1    INSTRUCTIONS

Participants in the public survey were first prompted to read a consent form detailing the tasks involved in the study and potential risks to participants, alongside estimates of payment for completing each study ($5 + bonus payments for forecasting performance in the introductory survey; $10 for completing the primary survey). In said consent form, the overall task was described as follows: "You will be asked to read some material, to follow some instructions, and to provide forecasts for future events." Then, participants were prompted to give their responses to 20 forecasting questions randomly selected from the 200-question subset of questions provided to the LLMs.

Participants were given a brief description of the task before each question:

> You are going to be predicting the probability of the answer to the question below being "Yes" (or "resolving positively").

For tasks with questions generated from data providers, forecasters were prompted to provide a probability forecast at multiple resolution dates.

The public survey was hosted on Qualtrics's (https://www.qualtrics.com/) survey software platform.

Participants in the superforecaster survey went through a similar initial experience before being moved into the group forecasting stage described in Section 4. Because of the expertise of this set of forecasters, superforecasters were also guaranteed a base payment of $1,000 for participating in the individual and group stages of the experiment as well as bonus payments for individual-level accuracy.

The superforecaster survey was hosted on Quorum (https://quorumapp.com/), a platform designed to host multi-stage forecasting tournaments.

### D.2    SCREENSHOTS

Figure 2 and Figure 3 show a few of the questions presented to participants in the public study.

*You are going to be predicting the probability of the answer to the question below being "Yes" (or "resolving positively").*

**Will a politician claim they lost a major election due to a "deepfake" image, video, or audio recording in a G20 country before 2025?**

"Deepfakes" or "Deep fakes" are synthetic media created with generative AI that imitate real media, but have people saying or doing things they did not actually do. This can be in the form of a static image, a short video, or an audio recording with no video.

In May 2022, a deepfake video of Elon Musk promoting a scam cryptocurrency went viral on social media, to which Elon replied on Twitter "Yikes. Def not me.". In June 2022, Google banned deepfake-generate AI from its Colab tool. In May 2023, a deepfake image showed an explosion at the US Pentagon, which was picked up by media and caused a temporary huge drop in stock markets.

As of May 2023, no major political scandal has erupted due to a particular deepfake successfully convincing a large block of voters that a politician did or said something they didn't actually do.

- **URL:** https://www.metaculus.com/questions/17180/deepfake-costs-election-before-2025/
- **Resolution Criteria:** Resolves to the outcome of the question found at https://www.metaculus.com/questions/17180/deepfake-costs-election-before-2025/.. This question resolves as **YES** if, by Dec 31, 2024, a politician in an election with >3M votes cast in a G20 country claims that they lost an election due to a deepfake video, image, or audio recording of them.
- **Last recorded value:** 29.0%
    - The community prediction.
- **Freeze date:** 2024-07-12 00:00:00
- **Market open date:** 2023-05-23 18:00:00
- **Market close date:** 2024-12-31 20:00:00

Prediction

Probability (0-100%)

Please write your rationale here. We encourage you to include the step-by-step process you used to come up with your forecast.

Figure 2: An example market-based question from the human survey.

*You are going to be predicting the probability of the answer to the question below being "Yes" (or "resolving positively").*

**According to Wikipedia, will Sarasadat Khademalsharieh have an Elo rating on the dates listed below that's at least 1% higher than on 2024-07-21?**

The International Chess Federation (FIDE) governs international chess competition. Each month, FIDE publishes the lists 'Top 100 Players', 'Top 100 Women', 'Top 100 Juniors' and 'Top 100 Girls' and rankings of countries according to the average rating of their top 10 players and top 10 female players.

To create the rankings, FIDE uses the Elo rating system, which is a method for calculating the relative skill levels of players in zero-sum games such as chess. The difference in the ratings between two players serves as a predictor of the outcome of a match. Two players with equal ratings who play against each other are expected to score an equal number of wins. A player whose rating is 100 points greater than their opponent's is expected to score 64%; if the difference is 200 points, then the expected score for the stronger player is 76%.

A player's Elo rating is a number which may change depending on the outcome of rated games played. After every game, the winning player takes points from the losing one. The difference between the ratings of the winner and loser determines the total number of points gained or lost after a game. If the higher-rated player wins, then only a few rating points will be taken from the lower-rated player. However, if the lower-rated player scores an upset win, many rating points will be transferred. The lower-rated player will also gain a few points from the higher rated player in the event of a draw. This means that this rating system is self-correcting. Players whose ratings are too low or too high should, in the long run, do better or worse correspondingly than the rating system predicts and thus gain or lose rating points until the ratings reflect their true playing strength.

Elo ratings are comparative only, and are valid only within the rating pool in which they were calculated, rather than being an absolute measure of a player's strength.

- **URL:** https://en.wikipedia.org/wiki/FIDE_rankings
- **Resolution Criteria:** Resolves to the value calculated from https://en.wikipedia.org/wiki/FIDE_rankings on the resolution date.
- **Last recorded value:** 2489.0
    - Sarasadat Khademalsharieh's ELO rating.
- **Freeze date:** 2024-07-12 00:00:00

*Please only enter numbers between 0 and 100 below, where 1 represents 1%, 10 represents 10%, etc...*

| | Prediction |
|---|---|
| Probability (0-100%) at 2024-07-28 | |
| Probability (0-100%) at 2024-08-20 | |

Figure 3: An example question generated from a data provider, in this case DBnomics, from the public survey. Two of eight forecast horizons for which we elicited forecasts are included above. The rationale text boxes (one for each forecast horizon) have also been excluded from the screenshot for brevity.

# E LLM "ENSEMBLE" BASELINE

## E.1 MODELS

To construct an ensemble baseline that includes diverse candidates, we evaluate models using the most recent forecasting dataset containing cross-domain questions with true resolutions from Halawi et al. (2024). We assess models from the following organizations: OpenAI, Mistral AI, Qwen, Google, Anthropic, and Meta. Using the same scratchpad prompting method from Halawi et al. (2024), we then select the top three models: GPT-4o, Gemini-1.5.Pro, Claude-3.5-Sonnet. See Table 16 for the results.

Table 16: Brier Scores from each LLM "crowd" candidate.

| Model | Scratchpad |
| --- | --- |
| **GPT-4o** | **0.207 (0.026)** |
| Llama-3-70b | 0.232 (0.020) |
| Mistral-Large | 0.233 (0.026) |
| Qwen-1.5-110b | 0.222 (0.025) |
| **Gemini-1.5-Pro** | **0.214 (0.025)** |
| **Claude-3.5-Sonnet** | **0.178 (0.025)** |
| GPT-4-0613 | 0.222 (0.009) |
| GPT-4-1106-Preview | 0.209 (0.012) |
| GPT-3.5-Turbo-1106 | 0.261 (0.010) |
| GPT-3.5-Turbo-Instruct | 0.257 (0.009) |
| Claude-2 | 0.219 (0.014) |
| Claude-2.1 | 0.215 (0.014) |
| Gemini-Pro | 0.230 (0.007) |
| Mistral-7B-Instruct | 0.243 (0.008) |
| Mistral-8x7B-Instruct | 0.238 (0.010) |
| Mixtral-8x7B-DPO | 0.248 (0.010) |
| Yi-34B-Chat | 0.241 (0.009) |
| Llama-2-7B | 0.264 (0.011) |
| Llama-2-13B | 0.268 (0.008) |
| Llama-2-70B | 0.282 (0.011) |

The highlights indicate the models we decide to use. Subscript numbers denote 2 standard errors. Above the divider line are the new models we evaluate, while below the divider line are the models evaluated by Halawi et al. (2024) on the same dataset using the same prompt.

## E.2 AGGREGATION METHODS

With the forecasts generated by the top 3 models selected in Section E.1, we then compare performance of 5 aggregation methods: Median, Trimmed Mean, Geometric Mean, and Geometric Mean of Log Odds (Satopää et al., 2014). See Table 17 for the results.

## E.3 OTHER DETAILS

**LLM Parameters.** We set the temperature to 0 and the max output token length to 2000.

# F PROMPTS

In this section, we present the following prompts: zero-shot (Figure 4), scratchpad (Figure 5), and three prompts (Figure 6, Figure 7, Figure 8) written by superforecasters from the Forecasting Research Institute used to construct the LLM "ensemble" baseline. Note that the scratchpad with information retrieval prompt is simply the scratchpad with an additional line "We have retrieved

Table 17: Brier Scores from each Aggregation Method.

| Aggregation Method | Brier Score |
|---|---|
| Median | 0.197 (0.024) |
| Trimmed Mean | 0.197 (0.024) |
| **Geometric Mean** | **0.194 (0.023)** |
| **Geometric Mean of Log Odds** | **0.194 (0.023)** |

The geometric mean and geometric mean of log odds are highlighted as they achieve the best performance. Subscript numbers denote 2 standard errors.

---

You are an expert superforecaster, familiar with the work of Tetlock and others. Make a prediction of the probability that the question will be resolved as true. You MUST give a probability estimate between 0 and 1 UNDER ALL CIRCUMSTANCES. If for some reason you can't answer, pick the base rate, but return a number between 0 and 1.
Question: {question}
Question Background: {background}
Resolution Criteria: {resolution_criteria}
Question close date: {close_date}
Output your answer (a number between 0 and 1) with an asterisk at the beginning and end of the decimal. Do not output anything else.
Answer:
{{ Insert answer here }}

---

Figure 4: Zero-shot Prompt from Halawi et al. (2024)

the following information for this question: {retrieved_info}" before the instructions begin. For combination questions, we slightly modify each of the above prompts by including the second question's information (Figure 9). Finally, we present the prompt used to determine whether a question is marked as valid or invalid in Figure 10.

## G    STATISTICAL DETAILS

While our pairwise bootstrapped $p$-values are precise, our statistical tests ignore one potentially important source of imprecision. We assume that each question is independent of the other questions. In a world where forecasting accuracy is quite correlated within a topic, or where most events are correlated, this may result in us overstating how confidently we can reject the equivalence of different models. We hope to explore this question in future work, but we are somewhat reassured by the fact that our gathering of forecasting questions from diverse domains and sources makes it unlikely that correlated questions would change our interpretation of these results in any meaningful way.

Our statistical tests are significantly more precise than the 95% confidence intervals for each model would imply because accuracy on each question is quite correlated across models. To understand this phenomenon, consider a hypothetical world where Model A outperforms Model B by a constant ($\epsilon$) on each question. No matter how close the performance of Models A and B are (how small $\epsilon$ is), and no matter how much variance there is in the accuracy of Model A across questions (which drives the 95% confidence interval surrounding Model A's accuracy), a pairwise bootstrap would show that Model A is more accurate than Model B. This is because forecasts of Model A and Model B are perfectly correlated.

## H    LEADERBOARDS: TOP 50

We show the best 50 performers on several leaderboards. In Table 18 we show the leaderboard for the human question set of 200 standard (non-combination) questions. Table 19 shows the leaderboard for the full LLM question set of 1,000 questions. Finally, in Table 20, we present the human leaderboard with combination questions included where humans provided forecasts on both components of the combination question. We derive human forecasts for these combination questions by treating each component of the combination question as independent.

```
Question:
{question}
Question Background:
{background}
Resolution Criteria:
{resolution_criteria}
Question close date: {close_date}
Instructions:
1. Given the above question, rephrase and expand it to help you do better answering. Maintain all information in the original question.
{{ Insert rephrased and expanded question.}}
2. Provide a few reasons why the answer might be no. Rate the strength of each reason.
{{ Insert your thoughts }}
3. Provide a few reasons why the answer might be yes. Rate the strength of each reason.
{{ Insert your thoughts }}
4. Aggregate your considerations. Think like a superforecaster (e.g. Nate Silver).
{{ Insert your aggregated considerations }}
5. Output an initial probability (prediction) given steps 1-4.
{{ Insert initial probability }}
6. Evaluate whether your calculated probability is excessively confident or not confident enough. Also, consider anything else that might
affect the forecast that you did not before consider.
{{ Insert your thoughts }}
7. Output your answer (a number between 0 and 1) with an asterisk at the beginning and end of the decimal. (For example, if there are n
resolution dates, you would output different *p* for each resolution date) Do not output anything else.
{{ Insert your answer }}
```

Figure 5: Scratchpad Prompt modified from Halawi et al. (2024)

## I   AGGREGATING HUMAN FORECASTS

We provide aggregated forecasts on benchmark questions from 500 members of the general public and 39 superforecasters, as described in Section 4:

- Public:                www.forecastbench.org/datasets/forecast_sets/2024-07-21/2024-07-21.ForecastBench.human_public.json
- Superforecasters:           www.forecastbench.org/datasets/forecast_sets/2024-07-21/2024-07-21.ForecastBench.human_super.json

As described in Section 4, members of the general public were recruited via Prolific and Facebook. First, participants completed an introductory survey designed to gather demographic information and evaluate performance on a few forecasting and comprehension tasks. Then, they were invited to take part in the main survey, featuring 20 random benchmark questions. Some participants were disqualified from participating in the main survey based on suspicious elements of their presurvey responses (e.g., answering questions rapidly, large numbers of submissions from the same IP address, etc.).

Superforecasters were invited to participate directly and were prompted to give forecasts for at least 20 questions from the benchmark (though, many chose to participate on additional questions). Superforecasters were also given access to other Superforecasters' forecasts and rationales and were allowed to comment on and update based on others' forecasts.

For each group, the median forecast for each forecasting question was taken to create the aggregated forecast sets. Code and individual forecasts are forthcoming once we can ensure that the text responses have been fully anonymized.

## J   REPRODUCE LLM FORECASTS

We evaluate 17 LLMs on our initial benchmark: GPT-3.5-Turbo-Instruct (Brown et al., 2020), GPT-4 (OpenAI, 2023), GPT-4o, Llama-2-70B (Touvron et al., 2023), Llama-3-7B, Llama-3-70B, Mistral-7B, Mistral-8x7B (Jiang et al., 2024a), Mistral-8x22B, Mistral-Large, Qwen1.5-110B-Chat, Claude-2.1 (Anthropic, 2023), Claude-3-Haiku, Claude-3.5-Sonnet, Claude-3-Opus (Anthropic, 2024), Gemini 1.5 Flash, and Gemini 1.5 Pro (Gemini Team, 2023).

To make inferences, we use APIs. For the GPT-suite, we use OpenAI's API; for Gemini-suite, we use Google's API; for Llama-suite, Mistral-7B, Mixtral-8x7B, Mixtral-8x22B and Qwen1.5-110B-Chat,

Question: {question}
Question Background: {background}
Resolution Criteria: {resolution_criteria}
Question close date: {close_date}
We have retrieved the following information for this question: {retrieved_info}
Instructions:
1. Given the above question, rephrase and expand it to help you do better answering. Maintain all information in the original question. {{ Insert rephrased and expanded question.}}
2. Let's start by coming up with a base-rate that could be helpful for forecasting this question. Come up with the best reference-class you can for this sort of event, and give a general base-rate that doesn't take into account factors unique to this question.
For instance, if the question were about the probability of a new technology being widely adopted within five years, you might look at historical data on the adoption rates of similar technologies as a reference class. Come up with a base-rate that could be relevant for this question.
The base-rate must be formatted as a clear probability (or number, in cases where you believe that to be more useful than a probability). For instance, imagine you are forecasting the probability that an incumbent president will be re-elected in an upcoming election in a hypothetical country. The past data shows that the incumbent has been elected 60% of the time.
Here, you would write 'The reference class I have chosen is the incumbent being elected. My base-rate is that the probability of the incumbent being re-elected is 0.6.' Give a justification for the base-rate, as well as a clear number.
Importantly, the base-rate should be as specific as it's possible to be without losing confidence that the number is correct. For instance, if you were forecasting on the probability of a hypothetical democratic country going to war in the next year, you should ideally produce a base-rate for a democratic country going to war in a given year, rather than simply thinking about a given country going to war.
{{ Insert your base rate }}
3. Now, let's think about factors specific to this question that may give us a good reason to deviate from the base-rate. Please give some reasons that the probability of this question resolving positively may be higher than the base rate. Please note specifically how they affect your forecast in terms of percentage point change. {{ Insert your thoughts }}
4. Now, let's think about reasons that the probability of this question resolving positively may be lower than the base rate. Please note specifically how they affect your forecast in terms of percentage point change. {{ Insert your thoughts }}
5. Consider any other factors that may affect the probability of this question resolving positively or negatively, that you have not already discussed in the previous two steps. {{ Insert your thoughts }}
6. Aggregate your considerations. Think like a superforecaster (e.g. Nate Silver). Give a ranking to each consideration based on how much you believe it ought to affect your forecast. {{ Insert your aggregated considerations }}
7. Are there any ways in which the question could resolve positively or negatively that you haven't considered yet, or that require some outside-the-box thinking? For example, if the question was 'Will Microsoft have a market capitalization of over $5tn by 2030', you might consider questions like:
How likely is it that Microsoft no longer exists in 2030? How likely is it that inflation erodes that value of the dollar as such that $5n is worth significantly less than it is today? How likely is it that there is a merger between Microsoft and another large company? How likely is it that Microsoft is broken up, as it is perceived to have monopoly power?
Here, we're thinking about things that are probably quite unlikely to happen, but should still be integrated into your forecast. Write up some possibilities and consider how they should be integrated into your final forecast. {{Insert your thoughts and considerations about how this should affect your forecast}}
8. Output an initial probability (prediction) given steps 1-7. {{ Insert initial probability. }}
9. Okay, now let's think about some other ways to consider how to forecast on this question. What would you say are the odds that if you could fast-forward and find out whether that statement is true or false, you would find out it's true? You must give an odds ratio. This odds ratio probably shouldn't be purely on the basis of the considerations in the previous steps, but you should think again about what you would expect to see if you could fast-forward into the future. If it helps, imagine that you're taking a bet. {{ Insert your odds ratio. }}
10. Given your rephrased statement from step 1, think of 2-3 statements that if you conditioned on their being TRUE, you would think it more or less likely that your statement would be TRUE as well. These statements must not DETERMINE OR BE LOGICALLY EQUIVALENT to the original statement. Be creative! {{ Insert 2 to 3 related statements. }}
11. For each of your related statements, give new odds of the original statement conditional on the related statement being TRUE. {{ For each related statement, insert new odds for the original statement. }}
12. Now consider each of your odds from the previous steps(steps 9 - 11), and come up with your all-things-considered odds ratio for the original statement. {{ Insert final odds for the original statement. }}
13. Now, convert that odds ratio to a probability between 0 and 1. {{Insert a probability}}
14. Now, consider the probability that you came up with in step 8, as well as the probability that you came up with in step 13. Which of these probabilities do you lean towards? How do you weigh them against one another? Write up your thoughts on which probability is more likely to be "correct", and then decide on a FINAL probability that will be used as your forecast. {{Insert your thoughts AND a final probability}}
15. Output your answer (a number between 0 and 1) with an asterisk at the beginning and end of the decimal. {{ Insert your answer }}

Figure 6: Superforecaster prompt 1

we use Together AI's API; for Mistral-Large, we use Mistral AI's API; and for the Claude-suite, we use Anthropic's API. To reproduce our results, people will need to gather these API keys.

We then record model predictions using several different methods: zero-shot prompting, scratchpad prompting, scratchpad prompting with retrieval augmentation, and scratchpad prompting with retrieval augmentation and aggregate human forecasts. For the scratchpad and information retrieval setting, we use the retrieval infrastructure from Halawi et al. (2024) and provide relevant news articles to the models in-context to reason about. Additionally, only models with a context window larger than 8,000 tokens were evaluated under the retrieval setting due to the inclusion of news articles in the prompt.

Question: {question}
Question Background: {background}
Resolution Criteria: {resolution_criteria}
Here's some related information from the news that I've collected for you: {retrieved_info}
Question close date: {close_date}
Instructions:
1. Rephrase the question as a statement about the future, e.g. you would rephrase "Will Biden be the U.S. president on January 1 2025?" as "Biden is the U.S. president on January 1 2025." {{ Insert question rephrased as a statement. }}
2. What would you say are the odds that if you could fast-forward and find out whether that statement is true or false, you would find out it's true? You must give an odds ratio. If it helps, imagine that you're taking a bet. {{ Insert your odds ratio. }}
3. Given your rephrased statement, think of 2-3 statements that if you conditioned on their being TRUE, you would think it more or less likely that your statement would be TRUE as well. These statements must not DETERMINE OR BE LOGICALLY EQUIVALENT to the original statement. Be creative! {{ Insert 2 to 3 related statements. }}
4. For each of your related statements, give new odds of the original statement conditional on the related statement being TRUE.insert new odds for the original statement. }}
5. Now consider each of your odds from the previous steps and come up with your all-things-considered odds ratio for the original statement. Output your answer (a number between 0 and 1) with an asterisk at the beginning and end of the decimal. {{ Insert final odds for the original statement. }}

Figure 7: Superforecaster prompt 2

Question: {question}
Question Background: {background}
Resolution Criteria: {resolution_criteria}
Relevant information we retrieved from news articles: {retrieved_info}
Question close date: {close_date}
Instructions:
1. Given the above question, rephrase and expand it to help you do better answering. Maintain all information in the original question. {{ Insert rephrased and expanded question.}}
2. Provide a few reasons why the answer might be no. Rate the strength of each reason. For now, ignore the evidence, ideas, and perspectives contained in the attached news articles. {{ Insert your thoughts }}
3. Provide a few reasons why the answer might be yes. Rate the strength of each reason. For now, ignore the evidence, ideas, and perspectives contained in the attached news articles. {{ Insert your thoughts }}
4. Aggregate the considerations you developed in the previous steps. Think like a superforecaster (e.g. Nate Silver). {{ Insert your aggregated considerations }}
5. Output an initial probability (prediction) given steps 1-4. {{ Insert initial probability. }}
6. Now, consider the perspectives, ideas, and evidence that was provided in the retrieved news articles. How should these affect your judgment of the probability of the question resolving positively? List all reasons why these news articles might increase the probability of the question resolving positively. {{Insert your thoughts}}
7. Now, let's focus on how the ideas, perspectives, and evidence provided in the news articles might decrease the probability of the question resolving positively. {{Insert your thoughts}}
8. Given what you've thought about in the previous two steps, update your probability from the initial probability you gave in step 5. {{Insert updated probability}}
9. Evaluate whether your calculated probability is excessively confident or not confident enough. Also, consider anything else that might affect the forecast that you did not before consider. {{ Insert your thoughts }}
10. Output your answer (a number between 0 and 1) with an asterisk at the beginning and end of the decimal. Do not output anything else. {{ Insert your answer }}

Figure 8: Superforecaster prompt 3

Additionally, to speed up the inference, we use multithreading with 50 workers, which requires a high rate limit and requests for better subscription plans from each source. However, one can run inference sequentially by setting it as 1 worker, but this requires longer time to generate all the baselines.

## J.1 ZERO-SHOT AND SCRATCHPAD BASELINES

**Prompts** We use the zero-shot and scratchpad prompts shown in Appendix F.

**Hyperparameters** For the zero-shot setting, we set the maximum output token length to 50 since we only request probabilistic forecasts. For the scratchpad prompt, we increase the maximum output token length to 1300 as it requires reasoning and probabilistic forecasts. We initially considered a high token length of 3000, but after observing that the maximum response length was around 1250, we settled on 1300 as the optimal maximum token length. In both cases, the model temperature is set to 0 to ensure stable outputs.

**How to Reproduce** To run zero-shot and scratchpad baselines, follow the steps below:

1. Insert all the necessary API keys in `src/helpers/keys.py`.

```
Question 1: {question_1}
Question 2: {question_2}
Question 1 Background: {background_1}
Question 2 Background: {background_2}
Question 1 Resolution Criteria: {resolution_criteria_1}
Question 2 Resolution Criteria: {resolution_criteria_2}
Question 1 Current value on {freeze_datetime_1}: {value_at_freeze_datetime_1}
Question 1 Value Explanation: {value_at_freeze_datetime_explanation_1}
Question 2 Current value on {freeze_datetime_1}: {value_at_freeze_datetime_2}
Question 2 Value Explanation: {value_at_freeze_datetime_explanation_2}
Here's some related information from the news that I've collected for Question 1: {retrieved_info_1}
Here's some related information from the news that I've collected for Question 2: {retrieved_info_2}
Question resolution date: {list_of_resolution_dates}
```

Figure 9: Combination prompt that includes information about both non-market questions. The instructions are truncated and can be supplemented with any of the prompts shown above.

```
I want to assess the quality of a forecast question.
Here is the forecast question:  {question}.
Please flag questions that don't seem appropriate by outputting "flag".  Otherwise, if it seems like a
reasonable question or if you're unsure, output "ok."
In general, poorly-defined questions, questions that are sexual in nature, questions that are too
personal,
questions about the death/life expectancy of an individual should be flagged or, more generally,
questions
that are not in the public interest should be flagged.  Geopolitical questions, questions about court
cases,
the entertainment industry, wars, public figures, and, more generally, questions in the public interest
should
be marked as "ok."
Examples of questions that should be flagged:
* "Will I finish my homework tonight?"
* "Metaculus party 2023"
* "Will Hell freeze over?"
* "Heads or tails?"
* "Will I get into MIT?"
* "Will this video reach 100k views by the EOD?"
* "If @Aella goes on the Whatever podcast, will she regret it?"
* "Daily coinflip"
* "Musk vs Zuckerberg:  Will either of them shit their pants on the mat?"
Examples of questions that should NOT be flagged:
* "Will Megan Markle and Prince Harry have a baby by the end of the year?"
* "Will the Brain Preservation Foundation's Large Mammal preservation prize be won by Feb 9th, 2017?"
* "Will there be more novel new drugs approved by the FDA in 2016 than in 2015?"
* "Will Israel invade Rafah in May 2024?"
* "Will Iraq return its ambassador to Iran in the next month?"
* "Tiger Woods Will Win Another PGA Tournament"
* "Will Dwayne Johnson win the 2024 US Presidential Election?"
* "Will Oppenheimer win best picture AND Bitcoin reach $70K AND Nintendo announce a new console by EOY
2024?"
* "Will anybody born before 2000 live to be 150?"
* "Will Taylor Swift get married before Bitcoin reaches $100K USD?"
* "Will Russia's total territory decrease by at least 20% before 2028?"
* "Will Donald Trump be jailed or incarcerated before 2030?"
* "If China invades Taiwan before 2035, will the US respond with military force?"
* "Will there be a tsunami that kills at least 50,000 people before 2030?"
* "Will there be a military conflict resulting in at least 50 deaths between the United States and China
in 2024?"
* "Will an AI system be reported to have successfully blackmailed someone for >$1000 by EOY 2028?"
* "Will Vladimir Putin declare Martial Law in at least 3/4 of Russia before 2025?"
Again, when in doubt, do NOT flag the question; mark it as "ok".
Your response should take the following structure:
Insert thinking:
{{ insert your concise thoughts here }}
Classification:
{{ insert "flag" or "ok"}}
```

Figure 10: Question validation prompt

2. Run the zero-shot and the scratchpad baselines in `src/base_eval/llm_baselines/ manager/main.py`.

## J.2    SCRATCHPAD WITH INFORMATION RETRIEVAL BASELINE

**Prompts** We use the same scratchpad prompt as scratchpad baseline with an additional line "We have retrieved the following information for this question: {retrieved_info}" before the instructions begin.

Table 18: Leaderboard: Human question set (top 50)

| Model | Organization | Information provided | Prompt | Brier Score ↓ | | | Confidence Interval | Pairwise p-value comparing to No. 1 | Pct. more accurate than No. 1 |
|---|---|---|---|---|---|---|---|---|---|
| | | | | Dataset ($N$=422) | Market ($N$=76) | Overall ($N$=498) | | | |
| Superforecaster median forecast | ForecastBench | – | – | 0.118 | 0.074 | 0.096 | [0.076, 0.116] | – | 0% |
| Public median forecast | ForecastBench | – | – | 0.153 | 0.089 | 0.121 | [0.101, 0.141] | <0.001 | 22% |
| Claude-3-5-Sonnet-20240620 | Anthropic | Freeze values | Scratchpad | 0.138 | 0.107 | 0.122 | [0.099, 0.146] | <0.001 | 31% |
| Claude-3-5-Sonnet-20240620 | Anthropic | News with freeze values | Scratchpad | 0.142 | 0.112 | 0.127 | [0.104, 0.150] | <0.001 | 29% |
| GPT-4-Turbo-2024-04-09 | OpenAI | Freeze values | Zero shot | 0.162 | 0.095 | 0.128 | [0.105, 0.151] | <0.001 | 32% |
| Claude-3-5-Sonnet-20240620 | Anthropic | Freeze values | Zero shot | 0.145 | 0.117 | 0.131 | [0.103, 0.159] | <0.001 | 31% |
| GPT-4 | OpenAI | Freeze values | Zero shot | 0.167 | 0.096 | 0.132 | [0.109, 0.155] | <0.001 | 31% |
| GPT-4o | OpenAI | News with freeze values | Scratchpad | 0.162 | 0.105 | 0.133 | [0.113, 0.154] | <0.001 | 25% |
| Claude-3-5-Sonnet-20240620 | Anthropic | – | Scratchpad | 0.138 | 0.133 | 0.136 | [0.113, 0.158] | <0.001 | 28% |
| GPT-4o | OpenAI | Freeze values | Scratchpad | 0.161 | 0.113 | 0.137 | [0.115, 0.158] | <0.001 | 27% |
| Claude-3-5-Sonnet-20240620 | Anthropic | News | Scratchpad | 0.142 | 0.137 | 0.139 | [0.117, 0.161] | <0.001 | 26% |
| Claude-3-Opus-20240229 | Anthropic | Freeze values | Zero shot | 0.163 | 0.115 | 0.139 | [0.114, 0.164] | <0.001 | 23% |
| Claude-3-5-Sonnet-20240620 | Anthropic | News | Superforecaster 2 | 0.158 | 0.123 | 0.140 | [0.120, 0.161] | <0.001 | 25% |
| GPT-4o | OpenAI | – | Scratchpad | 0.161 | 0.125 | 0.143 | [0.125, 0.160] | <0.001 | 24% |
| Claude-3-5-Sonnet-20240620 | Anthropic | News | Superforecaster 1 | 0.150 | 0.135 | 0.143 | [0.120, 0.166] | <0.001 | 25% |
| GPT-4o | OpenAI | News | Scratchpad | 0.162 | 0.126 | 0.144 | [0.122, 0.165] | <0.001 | 22% |
| Claude-3-Opus-20240229 | Anthropic | Freeze values | Scratchpad | 0.160 | 0.129 | 0.144 | [0.125, 0.163] | <0.001 | 23% |
| Mistral-Large-Latest | Mistral AI | Freeze values | Zero shot | 0.173 | 0.117 | 0.145 | [0.121, 0.169] | <0.001 | 23% |
| Gemini-1.5-Pro | Google | News with freeze values | Scratchpad | 0.163 | 0.130 | 0.146 | [0.127, 0.165] | <0.001 | 23% |
| Gemini-1.5-Pro | Google | – | Scratchpad | 0.162 | 0.131 | 0.147 | [0.129, 0.164] | <0.001 | 23% |
| Mistral-Large-Latest | Mistral AI | Freeze values | Scratchpad | 0.161 | 0.133 | 0.147 | [0.129, 0.165] | <0.001 | 22% |
| GPT-4-Turbo-2024-04-09 | OpenAI | – | Zero shot | 0.162 | 0.137 | 0.149 | [0.129, 0.170] | <0.001 | 23% |
| GPT-4-Turbo-2024-04-09 | OpenAI | Freeze values | Scratchpad | 0.176 | 0.123 | 0.150 | [0.126, 0.173] | <0.001 | 26% |
| GPT-4 | OpenAI | Freeze values | Scratchpad | 0.174 | 0.126 | 0.150 | [0.130, 0.170] | <0.001 | 21% |
| Gemini-1.5-Pro | Google | Freeze values | Scratchpad | 0.162 | 0.138 | 0.150 | [0.132, 0.169] | <0.001 | 22% |
| Claude-3-5-Sonnet-20240620 | Anthropic | – | Zero shot | 0.145 | 0.155 | 0.150 | [0.123, 0.177] | <0.001 | 24% |
| Gemini-1.5-Pro | Google | News | Scratchpad | 0.163 | 0.141 | 0.152 | [0.133, 0.171] | <0.001 | 22% |
| Claude-3-Opus-20240229 | Anthropic | News | Superforecaster 1 | 0.157 | 0.147 | 0.152 | [0.131, 0.174] | <0.001 | 22% |
| GPT-4-Turbo-2024-04-09 | OpenAI | News with freeze values | Scratchpad | 0.178 | 0.128 | 0.153 | [0.130, 0.175] | <0.001 | 25% |
| GPT-4o | OpenAI | Freeze values | Zero shot | 0.197 | 0.109 | 0.153 | [0.128, 0.178] | <0.001 | 26% |
| Llama-3-70b-Chat-Hf | Meta | Freeze values | Scratchpad | 0.191 | 0.116 | 0.153 | [0.135, 0.171] | <0.001 | 25% |
| Mixtral-8x22B-Instruct-V0.1 | Mistral AI | Freeze values | Scratchpad | 0.181 | 0.127 | 0.154 | [0.137, 0.172] | <0.001 | 21% |
| GPT-4-Turbo-2024-04-09 | OpenAI | – | Scratchpad | 0.176 | 0.133 | 0.154 | [0.137, 0.171] | <0.001 | 23% |
| Gemini-1.5-Pro | Google | Freeze values | Zero shot | 0.185 | 0.124 | 0.155 | [0.127, 0.182] | <0.001 | 25% |
| Claude-3-Opus-20240229 | Anthropic | – | Scratchpad | 0.160 | 0.149 | 0.155 | [0.136, 0.173] | <0.001 | 22% |
| GPT-4 | OpenAI | – | Scratchpad | 0.174 | 0.137 | 0.155 | [0.140, 0.171] | <0.001 | 18% |
| Qwen1.5-110B-Chat | Qwen | Freeze values | Zero shot | 0.197 | 0.115 | 0.156 | [0.134, 0.179] | <0.001 | 20% |
| Claude-2.1 | Anthropic | Freeze values | Scratchpad | 0.217 | 0.095 | 0.156 | [0.138, 0.175] | <0.001 | 24% |
| Gemini-1.5-Flash | Google | Freeze values | Zero shot | 0.191 | 0.125 | 0.158 | [0.130, 0.186] | <0.001 | 23% |
| Mixtral-8x22B-Instruct-V0.1 | Mistral AI | Freeze values | Zero shot | 0.185 | 0.131 | 0.158 | [0.131, 0.185] | <0.001 | 24% |
| Llama-3-70b-Chat-Hf | Meta | Freeze values | Zero shot | 0.194 | 0.124 | 0.159 | [0.133, 0.185] | <0.001 | 25% |
| GPT-4-Turbo-2024-04-09 | OpenAI | News | Scratchpad | 0.178 | 0.144 | 0.161 | [0.140, 0.182] | <0.001 | 24% |
| Imputed Forecaster | ForecastBench | – | – | 0.250 | 0.073 | 0.162 | [0.142, 0.181] | <0.001 | 26% |
| Qwen1.5-110B-Chat | Qwen | Freeze values | Scratchpad | 0.183 | 0.141 | 0.162 | [0.144, 0.180] | <0.001 | 18% |
| Claude-2.1 | Anthropic | – | Scratchpad | 0.217 | 0.109 | 0.163 | [0.143, 0.183] | <0.001 | 22% |
| Qwen1.5-110B-Chat | Qwen | News with freeze values | Scratchpad | 0.179 | 0.148 | 0.163 | [0.144, 0.183] | <0.001 | 22% |
| Claude-2.1 | Anthropic | Freeze values | Zero shot | 0.221 | 0.108 | 0.164 | [0.141, 0.188] | <0.001 | 27% |
| Mistral-Large-Latest | Mistral AI | – | Scratchpad | 0.161 | 0.168 | 0.165 | [0.146, 0.183] | <0.001 | 22% |
| Claude-3-5-Sonnet-20240620 | Anthropic | News | Scratchpad | 0.197 | 0.133 | 0.165 | [0.144, 0.186] | <0.001 | 19% |
| Claude-3-Opus-20240229 | Anthropic | – | Zero shot | 0.163 | 0.167 | 0.165 | [0.140, 0.191] | <0.001 | 18% |

*Notes:*

1. Shows performance on the 200 standard questions provided in the human question set at the 7-, 30-, 90-, and 180-day forecast horizons.
2. The full leaderboard is available at www.forecastbench.org. Online results are updated nightly, so may be slightly different than the version presented here.
3. For resolved market questions, forecasts are compared against ground truth while for unresolved market questions, they are compared to community aggregates.
4. The overall score is calculated as the average of the mean dataset Brier score and the mean market Brier score.
5. Pairwise $p$-value comparing to No. 1 (bootstrapped): The $p$-value calculated by bootstrapping the differences in overall score between each model and the best forecaster under the null hypothesis that there's no difference.
6. Pct. more accurate than No. 1: The percent of questions where this forecaster had a better overall score than the best forecaster.

**Information Retrieval** We use the same information retrieval system from Halawi et al. (2024). The pipeline consists of four steps: search query generation, news retrieval, relevance filtering and re-ranking, and text summarization. One must acquire a Newscatcher API key to implement the same retrieval method.

**Information Retrieval Hyperparameters** The hyperparameters were selected following the results in Section E.1 of Halawi et al. (2024), in which they used a greedy search approach to identify the optimal hyperparameters. We display the hyperparameters below:

**NUM_SEARCH_QUERY_KEYWORDS:** The number of keywords used in the search query. For our system, this is set to 6.

**MAX_WORDS_NEWSCATCHER:** The maximum number of words allowed in search queries for the NewsCatcher API. This is set to 5.

**MAX_WORDS_GNEWS:** The maximum number of words allowed in search queries for the Google News API. This is set to 8.

**SEARCH_QUERY_MODEL_NAME:** The name of the model used to generate search queries. We use gpt-4-1106-preview.

Table 19: Leaderboard: LLM question set (top 50)

| Model | Organization | Information provided | Prompt | Brier Score ↓ | | | Confidence Interval | Pairwise p-value comparing to No. 1 | Pct. more accurate than No. 1 |
| | | | | Dataset (N=5,492) | Market (N=897) | Overall (N=6,389) | | | |
|---|---|---|---|---|---|---|---|---|---|
| Claude-3-5-Sonnet-20240620 | Anthropic | Freeze values | Scratchpad | 0.169 | 0.078 | 0.123 | [0.117, 0.129] | – | 0% |
| GPT-4-Turbo-2024-04-09 | OpenAI | Freeze values | Scratchpad | 0.172 | 0.080 | 0.126 | [0.120, 0.132] | 0.096 | 43% |
| GPT-4o | OpenAI | Freeze values | Scratchpad | 0.186 | 0.069 | 0.128 | [0.122, 0.133] | <0.01 | 43% |
| Gemini-1.5-Pro | Google | Freeze values | Scratchpad | 0.162 | 0.106 | 0.134 | [0.128, 0.139] | <0.001 | 35% |
| GPT-4o | OpenAI | News with freeze values | Scratchpad | 0.190 | 0.084 | 0.137 | [0.131, 0.143] | <0.001 | 39% |
| Gemini-1.5-Pro | Google | News with freeze values | Scratchpad | 0.166 | 0.111 | 0.139 | [0.133, 0.144] | <0.001 | 34% |
| Claude-3-Opus-20240229 | Anthropic | Freeze values | Zero shot | 0.186 | 0.093 | 0.139 | [0.133, 0.146] | <0.001 | 41% |
| Qwen1.5-110B-Chat | Qwen | Freeze values | Scratchpad | 0.176 | 0.108 | 0.142 | [0.136, 0.148] | <0.001 | 30% |
| Claude-3-5-Sonnet-20240620 | Anthropic | News with freeze values | Scratchpad | 0.184 | 0.101 | 0.143 | [0.137, 0.149] | <0.001 | 32% |
| Claude-3-5-Sonnet-20240620 | Anthropic | Freeze values | Zero shot | 0.192 | 0.094 | 0.143 | [0.136, 0.150] | <0.001 | 42% |
| GPT-4-Turbo-2024-04-09 | OpenAI | – | Scratchpad | 0.172 | 0.115 | 0.143 | [0.138, 0.149] | <0.001 | 31% |
| GPT-4-Turbo-2024-04-09 | OpenAI | Freeze values | Zero shot | 0.204 | 0.084 | 0.144 | [0.137, 0.150] | <0.001 | 42% |
| Claude-3-5-Sonnet-20240620 | Anthropic | – | Scratchpad | 0.169 | 0.120 | 0.144 | [0.139, 0.150] | <0.001 | 10% |
| Gemini-1.5-Pro | Google | – | Scratchpad | 0.162 | 0.128 | 0.145 | [0.139, 0.151] | <0.001 | 32% |
| GPT-4 | OpenAI | Freeze values | Scratchpad | 0.194 | 0.100 | 0.147 | [0.141, 0.154] | <0.001 | 36% |
| Gemini-1.5-Pro | Google | News | Scratchpad | 0.166 | 0.129 | 0.147 | [0.141, 0.153] | <0.001 | 32% |
| Imputed Forecaster | ForecastBench | – | – | 0.250 | 0.048 | 0.149 | [0.145, 0.153] | <0.001 | 46% |
| GPT-4o | OpenAI | – | Scratchpad | 0.186 | 0.114 | 0.150 | [0.144, 0.156] | <0.001 | 31% |
| Gemini-1.5-Pro | Google | Freeze values | Zero shot | 0.217 | 0.083 | 0.150 | [0.144, 0.157] | <0.001 | 39% |
| GPT-4-Turbo-2024-04-09 | OpenAI | News with freeze values | Scratchpad | 0.211 | 0.091 | 0.151 | [0.145, 0.157] | <0.001 | 35% |
| GPT-4o | OpenAI | News | Scratchpad | 0.190 | 0.114 | 0.152 | [0.146, 0.158] | <0.001 | 31% |
| Claude-3-5-Sonnet-20240620 | Anthropic | News | Scratchpad | 0.184 | 0.124 | 0.154 | [0.148, 0.160] | <0.001 | 30% |
| GPT-4 | OpenAI | Freeze values | Zero shot | 0.222 | 0.087 | 0.154 | [0.148, 0.161] | <0.001 | 38% |
| Qwen1.5-110B-Chat | Qwen | – | Scratchpad | 0.176 | 0.134 | 0.155 | [0.150, 0.160] | <0.001 | 28% |
| LLM Crowd | ForecastBench | News | – | 0.242 | 0.068 | 0.155 | [0.151, 0.159] | <0.001 | 38% |
| Claude-3-Opus-20240229 | Anthropic | Freeze values | Scratchpad | 0.201 | 0.112 | 0.156 | [0.150, 0.162] | <0.001 | 27% |
| Mistral-Large-Latest | Mistral AI | Freeze values | Scratchpad | 0.199 | 0.115 | 0.157 | [0.151, 0.162] | <0.001 | 26% |
| Gemini-1.5-Pro | Google | – | Zero shot | 0.217 | 0.097 | 0.157 | [0.151, 0.163] | <0.001 | 37% |
| LLM Crowd | ForecastBench | News | – | 0.243 | 0.071 | 0.157 | [0.153, 0.161] | <0.001 | 37% |
| LLM Crowd | ForecastBench | News | – | 0.244 | 0.071 | 0.157 | [0.153, 0.161] | <0.001 | 37% |
| GPT-4 | OpenAI | – | Scratchpad | 0.194 | 0.121 | 0.158 | [0.153, 0.162] | <0.001 | 28% |
| Llama-3-70b-Chat-Hf | Meta | Freeze values | Zero shot | 0.215 | 0.101 | 0.158 | [0.151, 0.164] | <0.001 | 33% |
| Gemini-1.5-Pro | Google | News | Superforecaster 1 | 0.186 | 0.131 | 0.159 | [0.153, 0.165] | <0.001 | 31% |
| Claude-3-5-Sonnet-20240620 | Anthropic | News | Superforecaster 2 | 0.190 | 0.129 | 0.159 | [0.153, 0.165] | <0.001 | 29% |
| GPT-4-Turbo-2024-04-09 | OpenAI | – | Zero shot | 0.204 | 0.117 | 0.160 | [0.154, 0.167] | <0.001 | 32% |
| Gemini-1.5-Flash | Google | Freeze values | Scratchpad | 0.194 | 0.128 | 0.161 | [0.154, 0.168] | <0.001 | 32% |
| Claude-3-Opus-20240229 | Anthropic | – | Zero shot | 0.186 | 0.136 | 0.161 | [0.154, 0.168] | <0.001 | 35% |
| GPT-4-Turbo-2024-04-09 | OpenAI | News | Superforecaster 2 | 0.208 | 0.116 | 0.162 | [0.156, 0.167] | <0.001 | 28% |
| GPT-4-Turbo-2024-04-09 | OpenAI | News | Scratchpad | 0.211 | 0.114 | 0.163 | [0.157, 0.168] | <0.001 | 27% |
| Claude-3-5-Sonnet-20240620 | Anthropic | – | Zero shot | 0.192 | 0.134 | 0.163 | [0.156, 0.170] | <0.001 | 34% |
| Qwen1.5-110B-Chat | Qwen | News with freeze values | Scratchpad | 0.205 | 0.122 | 0.164 | [0.158, 0.170] | <0.001 | 26% |
| Llama-3-70b-Chat-Hf | Meta | Freeze values | Scratchpad | 0.221 | 0.108 | 0.164 | [0.159, 0.170] | <0.001 | 25% |
| Mistral-Large-Latest | Mistral AI | Freeze values | Zero shot | 0.208 | 0.122 | 0.165 | [0.157, 0.172] | <0.001 | 31% |
| Gemini-1.5-Flash | Google | Freeze values | Zero shot | 0.232 | 0.098 | 0.165 | [0.158, 0.173] | <0.001 | 40% |
| Mixtral-8x22B-Instruct-V0.1 | Mistral AI | Freeze values | Scratchpad | 0.210 | 0.121 | 0.165 | [0.159, 0.172] | <0.001 | 30% |
| GPT-4o | OpenAI | News | Superforecaster 3 | 0.211 | 0.124 | 0.168 | [0.162, 0.173] | <0.001 | 28% |
| Claude-2.1 | Anthropic | – | Scratchpad | 0.237 | 0.100 | 0.168 | [0.163, 0.174] | <0.001 | 38% |
| Gemini-1.5-Flash | Google | – | Scratchpad | 0.194 | 0.146 | 0.170 | [0.164, 0.176] | <0.001 | 28% |
| GPT-4o | OpenAI | Freeze values | Zero shot | 0.225 | 0.116 | 0.171 | [0.163, 0.178] | <0.001 | 37% |
| Claude-3-Opus-20240229 | Anthropic | – | Scratchpad | 0.201 | 0.141 | 0.171 | [0.165, 0.177] | <0.001 | 26% |

*Notes:*

1. Shows performance on the 1,000 (500 standard, 500 combination) questions in the LLM question set at the 7-, 30-, 90-, and 180-day forecast horizons.
2. The full leaderboard is available at www.forecastbench.org. Online results are updated nightly, so may be slightly different than the version presented here.
3. For resolved market questions, forecasts are compared against ground truth while for unresolved market questions, they are compared to community aggregates.
4. The overall score is calculated as the average of the mean dataset Brier score and the mean market Brier score.
5. Pairwise p-value comparing to No. 1 (bootstrapped): The p-value calculated by bootstrapping the differences in overall score between each model and the best forecaster under the null hypothesis that there's no difference.
6. Pct. more accurate than No. 1: The percent of questions where this forecaster had a better overall score than the best forecaster.

**SEARCH_QUERY_TEMPERATURE:** The temperature setting for the search query model, which controls the randomness of the output. We set this to 0.0 for deterministic outputs.

**SEARCH_QUERY_PROMPT_TEMPLATES:** The templates used to generate search queries. In our configuration, we use PROMPT_DICT["search_query"]["0"] and PROMPT_DICT["search_query"]["1"]. The exact search query can be found in search_query.py.

**NUM_ARTICLES_PER_QUERY:** The number of articles retrieved per search query. This is set to 10.

**SUMMARIZATION_MODEL_NAME:** The name of the model used for summarizing articles. We use gpt-3.5-turbo-1106.

**SUMMARIZATION_TEMPERATURE:** The temperature setting for the summarization model, which controls the randomness of the output. We set this to 0.2.

**SUMMARIZATION_PROMPT_TEMPLATE:** The template used for summarizing articles. In our configuration, we use PROMPT_DICT["summarization"]["9"]. The exact search query can be found in summarization.py.

Table 20: Leaderboard: Human question set with LLM question set combination questions (top 50)

| Model | Organization | Information provided | Prompt | Brier Score ↓ | | | Confidence Interval | Pairwise $p$-value comparing to No. 1 | Pct. more accurate than No. 1 |
|---|---|---|---|---|---|---|---|---|---|
| | | | | Dataset ($N$=1,754) | Market ($N$=296) | Overall ($N$=2,050) | | | |
| Superforecaster median forecast | ForecastBench | – | – | 0.091 | 0.062 | 0.076 | [0.067, 0.086] | – | 0% |
| Public median forecast | ForecastBench | – | – | 0.119 | 0.072 | 0.096 | [0.086, 0.105] | <0.001 | 23% |
| GPT-4o | OpenAI | Freeze values | Scratchpad | 0.175 | 0.085 | 0.130 | [0.119, 0.141] | <0.001 | 24% |
| Claude-3-5-Sonnet-20240620 | Anthropic | Freeze values | Scratchpad | 0.154 | 0.107 | 0.131 | [0.118, 0.143] | <0.001 | 24% |
| GPT-4-Turbo-2024-04-09 | OpenAI | Freeze values | Scratchpad | 0.164 | 0.101 | 0.133 | [0.121, 0.145] | <0.001 | 23% |
| GPT-4o | OpenAI | News with freeze values | Scratchpad | 0.171 | 0.104 | 0.137 | [0.125, 0.149] | <0.001 | 20% |
| Gemini-1.5-Pro | Google | Freeze values | Scratchpad | 0.152 | 0.130 | 0.141 | [0.130, 0.152] | <0.001 | 21% |
| Gemini-1.5-Pro | Google | News with freeze values | Scratchpad | 0.154 | 0.133 | 0.143 | [0.133, 0.154] | <0.001 | 21% |
| Claude-3-5-Sonnet-20240620 | Anthropic | News with freeze values | Scratchpad | 0.160 | 0.130 | 0.145 | [0.132, 0.158] | <0.001 | 20% |
| Claude-3-5-Sonnet-20240620 | Anthropic | Freeze values | Zero shot | 0.174 | 0.119 | 0.146 | [0.133, 0.160] | <0.001 | 22% |
| Gemini-1.5-Pro | Google | – | Scratchpad | 0.152 | 0.143 | 0.148 | [0.137, 0.158] | <0.001 | 20% |
| GPT-4-Turbo-2024-04-09 | OpenAI | – | Scratchpad | 0.164 | 0.132 | 0.148 | [0.138, 0.158] | <0.001 | 17% |
| Gemini-1.5-Pro | Google | News | Scratchpad | 0.154 | 0.143 | 0.148 | [0.137, 0.160] | <0.001 | 21% |
| Claude-3-5-Sonnet-20240620 | Anthropic | – | Scratchpad | 0.154 | 0.143 | 0.149 | [0.137, 0.160] | <0.001 | 20% |
| GPT-4o | OpenAI | – | Scratchpad | 0.175 | 0.122 | 0.149 | [0.138, 0.159] | <0.001 | 19% |
| Claude-3-Opus-20240229 | Anthropic | Freeze values | Zero shot | 0.173 | 0.124 | 0.149 | [0.135, 0.162] | <0.001 | 21% |
| GPT-4o | OpenAI | News | Scratchpad | 0.171 | 0.127 | 0.149 | [0.138, 0.160] | <0.001 | 18% |
| GPT-4-Turbo-2024-04-09 | OpenAI | Freeze values | Zero shot | 0.200 | 0.100 | 0.150 | [0.138, 0.162] | <0.001 | 24% |
| Qwen1.5-110B-Chat | Qwen | Freeze values | Scratchpad | 0.171 | 0.131 | 0.151 | [0.140, 0.162] | <0.001 | 16% |
| Claude-3-5-Sonnet-20240620 | Anthropic | News | Scratchpad | 0.160 | 0.149 | 0.154 | [0.143, 0.166] | <0.001 | 19% |
| Imputed Forecaster | ForecastBench | – | – | 0.250 | 0.059 | 0.155 | [0.147, 0.163] | <0.001 | 22% |
| GPT-4 | OpenAI | Freeze values | Zero shot | 0.213 | 0.099 | 0.156 | [0.144, 0.168] | <0.001 | 21% |
| Gemini-1.5-Pro | Google | Freeze values | Zero shot | 0.205 | 0.110 | 0.157 | [0.144, 0.171] | <0.001 | 20% |
| Claude-3-5-Sonnet-20240620 | Anthropic | News | Superforecaster 2 | 0.167 | 0.149 | 0.158 | [0.146, 0.169] | <0.001 | 17% |
| GPT-4 | OpenAI | Freeze values | Scratchpad | 0.190 | 0.125 | 0.158 | [0.145, 0.171] | <0.001 | 19% |
| Claude-3-Opus-20240229 | Anthropic | Freeze values | Scratchpad | 0.185 | 0.134 | 0.159 | [0.148, 0.171] | <0.001 | 18% |
| LLM Crowd | ForecastBench | News | – | 0.241 | 0.080 | 0.161 | [0.153, 0.168] | <0.001 | 18% |
| GPT-4-Turbo-2024-04-09 | OpenAI | News with freeze values | Scratchpad | 0.209 | 0.114 | 0.161 | [0.149, 0.173] | <0.001 | 20% |
| LLM Crowd | ForecastBench | News | – | 0.242 | 0.083 | 0.162 | [0.155, 0.170] | <0.001 | 18% |
| LLM Crowd | ForecastBench | News | – | 0.243 | 0.082 | 0.162 | [0.155, 0.170] | <0.001 | 18% |
| Gemini-1.5-Pro | Google | News | Superforecaster 1 | 0.176 | 0.151 | 0.164 | [0.153, 0.175] | <0.001 | 19% |
| Mistral-Large-Latest | Mistral AI | Freeze values | Scratchpad | 0.185 | 0.143 | 0.164 | [0.154, 0.175] | <0.001 | 16% |
| GPT-4 | OpenAI | – | Scratchpad | 0.190 | 0.140 | 0.165 | [0.156, 0.174] | <0.001 | 15% |
| Qwen1.5-110B-Chat | Qwen | – | Scratchpad | 0.171 | 0.161 | 0.166 | [0.156, 0.175] | <0.001 | 15% |
| Gemini-1.5-Pro | Google | – | Zero shot | 0.205 | 0.128 | 0.167 | [0.154, 0.179] | <0.001 | 19% |
| Gemini-1.5-Flash | Google | Freeze values | Scratchpad | 0.179 | 0.154 | 0.167 | [0.153, 0.180] | <0.001 | 18% |
| Claude-2.1 | Anthropic | – | Scratchpad | 0.228 | 0.105 | 0.167 | [0.157, 0.177] | <0.001 | 20% |
| Llama-3-70b-Chat-Hf | Meta | Freeze values | Zero shot | 0.205 | 0.132 | 0.168 | [0.155, 0.182] | <0.001 | 19% |
| Llama-3-70b-Chat-Hf | Meta | Freeze values | Scratchpad | 0.208 | 0.129 | 0.169 | [0.158, 0.179] | <0.001 | 17% |
| Claude-3-Opus-20240229 | Anthropic | – | Zero shot | 0.173 | 0.165 | 0.169 | [0.156, 0.183] | <0.001 | 18% |
| Gemini-1.5-Flash | Google | Freeze values | Zero shot | 0.217 | 0.122 | 0.169 | [0.155, 0.183] | <0.001 | 23% |
| GPT-4-Turbo-2024-04-09 | OpenAI | – | Zero shot | 0.200 | 0.139 | 0.169 | [0.157, 0.182] | <0.001 | 19% |
| GPT-4-Turbo-2024-04-09 | OpenAI | News | Scratchpad | 0.209 | 0.131 | 0.170 | [0.159, 0.180] | <0.001 | 17% |
| Gemini-1.5-Flash | Google | – | Scratchpad | 0.179 | 0.161 | 0.170 | [0.159, 0.181] | <0.001 | 16% |
| GPT-4-Turbo-2024-04-09 | OpenAI | News | Superforecaster 2 | 0.202 | 0.139 | 0.170 | [0.160, 0.181] | <0.001 | 17% |
| Qwen1.5-110B-Chat | Qwen | News with freeze values | Scratchpad | 0.198 | 0.146 | 0.172 | [0.161, 0.183] | <0.001 | 16% |
| Claude-3-5-Sonnet-20240620 | Anthropic | – | Zero shot | 0.174 | 0.171 | 0.172 | [0.158, 0.187] | <0.001 | 17% |
| Mistral-Large-Latest | Mistral AI | – | Zero shot | 0.203 | 0.145 | 0.174 | [0.160, 0.188] | <0.001 | 19% |
| Claude-2.1 | Anthropic | Freeze values | Scratchpad | 0.228 | 0.120 | 0.174 | [0.162, 0.186] | <0.001 | 20% |
| GPT-4o | OpenAI | News | Superforecaster 3 | 0.206 | 0.145 | 0.175 | [0.165, 0.186] | <0.001 | 16% |

*Notes:*

1. This shows performance on all 200 standard questions from the human question set *plus* those combination questions from the LLM question set where humans provided forecasts on both components ($Q1$ and $Q2$). LLM scores are only for this combined question set. Human forecasts for combination questions are generated from their forecasts on the component questions by assuming independence (which is not always the case, putting humans at a disadvantage). Evaluated at the 7-, 30-, 90-, and 180-day forecast horizons.
2. The full leaderboard is available at www.forecastbench.org. Online results are updated nightly, so may be slightly different than the version presented here.
3. For resolved market questions, forecasts are compared against ground truth while for unresolved market questions, they are compared to community aggregates.
4. The overall score is calculated as the average of the mean dataset Brier score and the mean market Brier score.
5. Pairwise $p$-value comparing to No. 1 (bootstrapped): The $p$-value calculated by bootstrapping the differences in overall score between each model and the best forecaster under the null hypothesis that there's no difference.
6. Pct. more accurate than No. 1: The percent of questions where this forecaster had a better overall score than the best forecaster.

**NUM_SUMMARIES_THRESHOLD:** The threshold number of summaries to generate. This is set to 10.

**PRE_FILTER_WITH_EMBEDDING:** A boolean flag indicating whether to pre-filter articles using embeddings. This is set to `True`.

**PRE_FILTER_WITH_EMBEDDING_THRESHOLD:** The threshold for pre-filtering articles using embeddings. This is set to 0.32.

**RANKING_MODEL_NAME:** The name of the model used for ranking articles. We use `gpt-3.5-turbo-1106`.

**RANKING_TEMPERATURE:** The temperature setting for the ranking model, which controls the randomness of the output. We set this to 0.0 for deterministic outputs.

**RANKING_PROMPT_TEMPLATE:** The template used for ranking articles. In our configuration, we use `PROMPT_DICT["ranking"]["0"]`.

**RANKING_RELEVANCE_THRESHOLD:** The relevance threshold for ranking articles. This is set to 4.

**RANKING_COSINE_SIMILARITY_THRESHOLD:** The cosine similarity threshold used in ranking. This is set to 0.5.

**SORT_BY:** The criterion used to sort articles. We sort by `date`.

**RANKING_METHOD:** The method used for ranking articles. We use `llm-rating`.

**RANKING_METHOD_LLM:** The specific method for ranking articles using the LLM. We use `title_250_tokens`, meaning ranking articles based on their titles and the first 250 tokens.

**NUM_SUMMARIES_THRESHOLD:** The threshold number of summaries to generate for final output. This is set to 20.

**EXTRACT_BACKGROUND_URLS:** A boolean flag indicating whether to extract background URLs from the articles. This is set to `True`.

**Inference Hyperparameters:** We set the maximum output token length to 2000 to accommodate reasoning and probabilistic forecasts. We set the model temperature to 0 to ensure stable outputs.

**How to reproduce** To run the Scratchpad with Information Retrieval baseline, follow these steps:

1. To run the information retrieval part:
   (a) Insert all the necessary API keys in `llm_retrieval/forecasting-llm-retrieval/config/keys.py`. Specifically, add the Newscatcher and OpenAI API keys.
   (b) Run `llm_retrieval/notebooks/retrieval_cache.ipynb`.
   (c) Save all the retrieved news under a folder called `news`.
2. To run the scratchpad with the information retrieval baseline:
   (a) Insert all the necessary API keys in `src/helpers/constants.py`.
   (b) Place the "news" folder in the same directory as `src/base_eval/all_recurrent_llm_baselines/main.py`.
   (c) Run `src/base_eval/all_recurrent_llm_baselines/main.py`.

### J.3 LLM "ENSEMBLE" BASELINE

To produce the LLM ensemble forecast, we query three models: GPT-4o, Claude-3.5-Sonnet, and Gemini-1.5-Pro. We use three prompts crafted by superforecasters who were given explicit instructions to write prompts that would help an LLM produce accurate forecasts This results in $3 \times 3 = 9$ forecasts per question. We then show 3 LLM crowd baselines using the median, geometric mean, and geometric mean of log odds.

**Prompts** We use the 3 superforecaster-written prompts shown in the appendix of our paper as Superforecaster Prompt 1-3.

**Inference hyperparameters** We set the maximum output token length to 2000 to accommodate reasoning and probabilistic forecasts. We initially considered a high token length of 3000, but after observing that the maximum response length was around 1950, we finalized 2000 as the optimal maximum token length. We set the model temperature to 0 to ensure stable outputs.

**How to reproduce** To run LLM "Ensemble" baseline, follow the below steps:

1. Insert all the necessary API keys in `src/helpers/constants.py`.
2. Place the "news" folder in the same directory as `src/base_eval/llm_crowd/notebook.ipynb`.
3. Run `src/base_eval/llm_crowd/notebook.ipynb`.

## K  REPRODUCE RESOLUTION AND LEADERBOARD

Given the forecast files output from Appendix I and Appendix J, the forecasts can be resolved and the leaderboard created as outlined below, after first having downloaded the benchmark codebase.

The Google Cloud Run Job in `src/resolve_forecasts/main.py` resolves all forecasts on the questions from the question set in Section B.1. To do this, it depends on:

- the forecast files provided in Section B.2.1 and Section B.2.3;

- the complete resolution files from our Question Bank on GCP Cloud Storage, which we cannot distribute freely because some providers do not allow us to distribute their data directly, rather only modifications of their data. However, the code to create these resolution files is provided under `src/questions` and can be created given the API keys to the data sources.

Having resolved the forecasts for the day, either to ground truth if it was a forecast on a dataset question, or the resolution value or market value for market questions, we can now create the leaderboard. To do this, we use the Google Cloud Run Job defined in `src/leaderboard/main.py`.

## L  GENERAL PUBLIC SURVEY DEMOGRAPHICS

We collected demographic information from the 500 human forecasters in the general public survey. Summaries of participants' age, gender, ethnicity, and country of residence are shown in the tables below.

Table 21: Age Distribution

| Age | Percentage |
| --- | --- |
| 18–24 years old | 32.0% |
| 25–34 years old | 43.4% |
| 35–44 years old | 14.4% |
| 45–54 years old | 5.4% |
| Over 55 | 4.8% |

Table 22: Gender Distribution

| Gender | Percentage |
| --- | --- |
| Male | 53.4% |
| Female | 46.2% |
| Prefer not to say | 0.4% |

Table 23: Ethnicity Distribution

| Ethnicity | Percentage |
| --- | --- |
| White | 48.6% |
| Black | 33.4% |
| Mixed | 8.6% |
| Asian | 4.4% |
| Other | 3.4% |
| Prefer not to say | 1.6% |

We did not collect similar demographic information from the superforecasters participating in the study, but are reasonably certain that the superforecasters in this study are roughly representative of superforecasters as a whole. Describing forecasters previously recruited by Good Judgment Project, Mellers et al. (2015) noted that they "tended to be men (83%) and U.S. citizens (74%), with an average age of 40 years."

## M  PERFORMANCE BREAKDOWN

Tables Table 25 and Table 26 show the performance of the top LLM—Claude 3.5 Sonnet (using the Scratchpad prompt with freeze values)—compared with Superforecasters, evaluated by forecast category and horizon.

Table 24: Country of Residence Distribution

| Country | Percentage |
|---|---|
| South Africa | 31.2% |
| United States | 15.4% |
| Poland | 9.0% |
| Portugal | 7.8% |
| United Kingdom | 4.6% |
| Mexico | 4.2% |
| Chile | 3.8% |
| Italy | 3.4% |
| Greece | 3.0% |
| Hungary | 3.0% |
| (25 others) | 14.6% |

In Table 25, we see that Claude 3.5 Sonnet slightly outperformed superforecasters on Environment & Energy questions.

As a reminder, we ask for forecasts on dataset questions at 8 diffreent forecast horizons, as explained in Section 3.2. Here, we breakdown performance at the forecast horizons that have resolved by the publication date: 7 days, 30 days, 90 days, and 180 days. We see that superforecasters outperformed Claude 3.5 Sonnet at all forecast horizons, except for the 90-day horizon where Claude 3.5 Sonnet performed better.

Table 25: Brier score by category

| Category | N | Claude 3.5 Sonnet | Superforecaster | Difference |
|---|---|---|---|---|
| Arts & Recreation | 8 | 0.102 | 0.066 | 0.036 |
| Economics & Business | 181 | 0.224 | 0.134 | 0.090 |
| Environment & Energy | 81 | 0.126 | 0.141 | -0.015 |
| Healthcare & Biology | 48 | 0.003 | 0.001 | 0.002 |
| Politics & Governance | 17 | 0.066 | 0.022 | 0.044 |
| Science & Tech | 14 | 0.207 | 0.175 | 0.032 |
| Security & Defense | 95 | 0.056 | 0.030 | 0.026 |
| Sports | 54 | 0.058 | 0.034 | 0.024 |

Table 26: Brier score by forecast horizon

| Forecast Horizon | N | Claude 3.5 Sonnet | Superforecaster | Difference |
|---|---|---|---|---|
| 7-day | 106 | 0.093 | 0.076 | 0.017 |
| 30-day | 112 | 0.163 | 0.121 | 0.042 |
| 90-day | 110 | 0.151 | 0.172 | -0.021 |
| 180-day | 169 | 0.110 | 0.082 | 0.028 |

