# OpenReview forum: "ForecastBench: A Dynamic Benchmark of AI Forecasting Capabilities"
_ICLR.cc/2025/Conference — ICLR 2025 Poster_

### Official Review · Reviewer_1r9B · 2024-11-02

**Soundness:** 2
**Presentation:** 3
**Contribution:** 3
**Rating:** 5
**Confidence:** 4

**Summary:**

The paper introduces ForecastBench, a dynamic benchmark for evaluating the forecasting capabilities of both human and machine learning systems. This benchmark continuously updates with new forecasting questions, addressing the limitations of static benchmarks. The paper includes comprehensive experiments comparing the performance of various LLMs and human forecasters.

**Strengths:**

Interesting paper and relevant problem statement. Human annotated data generation in a streamlined manner is a challenge and the methodology proposed in the work can be used for updating the evaluation set for LLMs with time.
The work also discusses the evaluation on complex questions comprising of simple questions.

Paper is well-written.

**Weaknesses:**

Overall concerns-
Forecasting based on questions is a highly subjective problem. The questions and their respective responses could vary depending on the demographics, education background, understanding of the domain, etc of the forecasters. Does the authors have any openly accessible user-study report highlighting the biases and how they are being addressed. This follows for both - human forecasters as well as "superforecasters"

I'm wondering how does the 1000 questions designed for evaluation covers forecasting of an open world set? Or is it only targeted to a few domains?

The question examples shown in Figure 4&5 seem at a very high-resolution. Although, it is appreciated that the source of the questions is provided, but "is 2024 going to be hotter than 2023" - What parameter is being considered here for comparison? is it mean temperature? Can LLMs infer these for a new undefined concept?

Methodology relies on online links and APIs, thus limited in its coverage of questions.

Reproducibility is missing; validation is difficult.

User -study report should be provided publicly. What's the demographics of the human forecasters?

Reliability and risks associated with the methodology - Is stress test performed on the methodology for use in real-world? What are the metrics?

**Questions:**

Questions in Weakness Section.

---

> ### Author Response · Authors · 2024-11-24
>
> Thank you for your valuable feedback\! We address your questions and concerns below.
>
> ---
>
> ### Weaknesses
>
> **\> Overall concerns- Forecasting based on questions is a highly subjective problem. The questions and their respective responses could vary depending on the demographics, education background, understanding of the domain, etc of the forecasters. Does the authors have any openly accessible user-study report highlighting the biases and how they are being addressed. This follows for both \- human forecasters as well as "superforecasters"**
>
> This is a valid point. We have updated our draft to include Appendix L describing the demographic information of our survey respondents, mentioned the potential issues we see regarding the impact of demographics on both questions and forecasts, and referenced this section from a footnote in the main text.
>
> **\> I'm wondering how does the 1000 questions designed for evaluation covers forecasting of an open world set? Or is it only targeted to a few domains?**
>
> Of the 500 standard questions in the LLM dataset, 250 come from forecasting websites that can cover any subject matter. New questions come in daily and are informed by world events and hence are diverse and cover a broad set of domains.
>
> The other 250 questions come from datasets. We’ve selected diverse data sets (ACLED, FRED, Wikipedia, …) to ensure coverage of many different domains.
>
> Indeed, one reason we categorize the questions in the ForecastBench question bank by topic is to achieve a broad representation of domains. When sampling from the question bank to generate the question set, we sample questions by category within each source to ensure topic diversity.
>
> A detailed breakdown of categories, with the number of questions available in the question bank per category, is available in Table 15 in Appendix C.2.
>
> **\> The question examples shown in Figure 4&5 seem at a very high-resolution. Although, it is appreciated that the source of the questions is provided, but "is 2024 going to be hotter than 2023" \- What parameter is being considered here for comparison? is it mean temperature? Can LLMs infer these for a new undefined concept?**
>
> Before responding to the query about this question, we want to note that the vast majority of our questions have clear resolution criteria. Though there are a few exceptions (as you note in your question), we have closely examined our dataset and ensured they are a very small minority of questions and do not impact our mainline results.
>
> Regarding this specific question, you correctly point out that the resolution criteria is not clear. That has not prevented human forecasters from converging on a response (the crowd forecast as of 2024-11-13 is 98.9%, when the question was asked the crowd forecast was 76%). Indeed, in an active market, the crowd forecast should incorporate all pertinent information (information about recent events, short-run changes in weather, and even the resolution criteria). If a question has ambiguous resolution criteria, that should be reflected in the crowd forecast.
>
> We have replaced this question in the text with a more representative question. You can see the updated question in Figure 2 of Appendix D.2. And we plan to impose additional filters to catch/remove questions like this in the future.
>
> **\> Methodology relies on online links and APIs, thus limited in its coverage of questions.**
>
> To ensure coverage by source and topic, we specifically draw questions from a broad range of sources, including databases of geopolitical events, economic indicators, weather time series, sports records, and current events. We therefore think that our benchmark contains the broadest coverage of any set of forecasting questions constructed in the literature, and we plan to continue to expand the coverage of the question set over time.
>
> To more specifically answer one of your points, the ‘online links’ we rely on are themselves forecasting platforms that write and curate thousands of forecasting questions, updating the available questions and topics regularly. By relying on these sources, we ensure coverage of key real-world events in our benchmark.
>
> The tradeoff here is that we have a dynamic, ever-expanding source of questions that changes with current events and remains relevant. This automatically-updated system requires minimal maintenance.
>
> Currently, we draw from a large enough set of online databases and APIs to ensure robustness of our benchmark to API instability and downtime. If you have suggestions for how to improve or modify our setup to ensure broader coverage of domains, we’re happy to address them.

---

> > ### Author Response · Authors · 2024-11-24
> >
> > **\> Reproducibility is missing; validation is difficult.**
> >
> > Our results are fully reproducible.
> >
> > The code is publicly available on Github (MIT license), as are the question sets, resolution sets, and forecast sets used for generating the leaderboard (CC-BY-SA license).
> >
> > Further, our leaderboard will continue to be replicable. To achieve that, we are maintaining a ForecastBench dataset repository that is updated nightly with the data necessary to recreate the leaderboard. We store the resulting leaderboard in the repository as well. Given this data and our source code, one can recreate the leaderboard on any given date.
> >
> > To reproduce LLM forecasts, see the exposition in Appendices I, J, and K.
> >
> > One can thus validate our findings with our code, the datasets we provide, a GCP account, and appropriate model API access.
> >
> > **\> User \-study report should be provided publicly. What's the demographics of the human forecasters?**
> >
> > Thank you for the question, we have updated the draft to include the demographic information for the public forecasters in Appendix L. We did not collect demographic information for the superforecasters but they are representative of superforecasters as a whole, who have been previously described as tending “to be men (83%) and U.S. citizens (74%), with an average age of 40 years” according to Mellers et al. (2015).
> >
> > ---
> >
> > Please let us know if you have any other questions or concerns we can address. If not, we hope that you will consider increasing your rating.

---

> > > ### Comment · Reviewer_1r9B · 2024-11-26
> > >
> > > Thank you for your detailed responses. This paper does not propose new advanced machine learning methodology or demonstrate novel ML techniques. Based on this, I would maintain my score. This is a good application paper but not for ICLR, maybe for datasets and benchmark tracks.

---

> > > > ### Author Response · Authors · 2024-11-27
> > > >
> > > > Thank you for your feedback. We wanted to clarify an important point. ICLR does not have a separate track for datasets and benchmarks. Instead, the 2025 call for papers explicitly requests submissions across many applied domains, including “datasets and benchmarks” and it embraces a broad spectrum of topics, as stated on its official website: "A non-exhaustive list of relevant topics [includes]:... datasets and benchmarks" https://iclr.cc/Conferences/2025/CallForPapers
> > > >
> > > > Furthermore, in that call for papers, ICLR organizers clearly state that
> > > >
> > > > > We consider a broad range of subject areas including… applications in vision, audio, speech, language, music, robotics, games, healthcare, biology, sustainability, economics, ethical considerations in ML, and others.
> > > >
> > > > ForecastBench has clear applications to economics and other fields in that it explores the usefulness and accuracy of ML-based systems, built using LLMs, to forecast important economic and geopolitical indicators that are often inputs into policymaking (inflation expectations, the likelihood of war between two countries,...).
> > > >
> > > > The language in the call for papers makes it clear that impactful contributions to ICLR are not limited to “advanced machine learning methodology” or “novel ML techniques” but also include advances in datasets, benchmarks, and application-driven insights.

---

> > > > > ### Author Response · Authors · 2024-11-27
> > > > >
> > > > > At a practical level, ForecastBench is a useful benchmark of the forecasting abilities of AI-based systems. But at a conceptual level, we believe our paper makes several other contributions including, most importantly, being one of the first automatically updating dynamic benchmarks, capable of systematically addressing significant limitations in existing static benchmarks for AI-based systems. By enabling real-time, contamination-free evaluations of the quality of AI-based systems, ForecastBench sets a new standard for reliability and relevance in benchmarking practices. For example, Zhang et al. https://arxiv.org/pdf/2405.00332 showed that on fresh GSM8K questions, model performance dropped by 13% for some families of models. This is no different in forecasting; a recent study claiming that LLMs are superhuman at forecasting (‘LLMs are Superhuman Forecasters’, by Phan et al.)) has been removed from arxiv after other researchers showed that the claims did not hold up when evaluated on a new set of questions. Here, since our benchmark is dynamic -- automatically adding fresh forecasting questions about events that have not happened yet -- we avoid potential leakage problems. We hope we set an example for the field to follow so we can more accurately assess models. These contributions fall squarely under the scope of "datasets and benchmarks" and "applications," particularly in some of the fields listed in the call for papers.
> > > > >
> > > > > In addition to its practical use as a benchmark, the recently revised and uploaded version of our paper adds several new pieces, including (1) a discussion of the general importance of the task that is at the core of our benchmark and (2) the inclusion of new analyses on scaling trends and on the relationship between model performance on forecasting questions and established benchmarks and training compute (in Section 5.2). We go into more detail on those contributions, below.
> > > > >
> > > > > A key topic of research is whether LLMs have achieved human parity in general reasoning, and if they have not, when or if they will. Forecasting is a useful testbed of LLM reasoning ability, just like math and coding. To produce an accurate forecast, a person or AI system must synthesize information, ideally from both live and preexisting knowledge, reason, avoid overconfidence, guard against biases, combine information, and quantify beliefs. Among the 17 models we test (across the 7 baselines outlined in Section 5.1), only Claude-3.5 Sonnet performed as well as the general public (Brier score of 0.114), and both Claude-3.5 and the general public perform significantly worse than superforecaster-level (Brier score of 0.092, lower is better). Our benchmark thus provides valuable insights into the capability of AI systems.
> > > > >
> > > > > Lastly, we recently analyzed the scaling trends of LLMs to observe how forecasting performance scales with more training compute. Figure 1 shows a negative log-linear relationship between training compute and forecasting accuracy (Brier score). This relationship was marginally statistically insignificant ($r = -0.64$, $p = 0.06$) because of the small number of tested models (only 9 models we tested had reliable estimates of training compute). But, we use this pattern to project out the log-linear relationship, finding that if performance continues to improve at these rates, LLMs could match superforecaster performance when training compute approaches $1 \times 10^{27}$ FLOP, though there is a large confidence interval (bootstrapped 95% CI: [$7.08 \times 10^{26}$,$3.64 \times 10^{30}$].
> > > > >
> > > > > We also analyzed the relationship between Chatbot Arena scores (see Chiang et al., 2024) and Brier scores, for which we could compare all 17 models tested. Here we found a strong and highly statistically significant relationship ($r = -0.69$, $p = 0.002$), showing that models with higher Arena scores also tend to produce more accurate forecasts. The calculated linear relationship implies that LLMs will match superforecaster performance when the Arena score approaches 1398 (bootstrapped 95% CI: [1335,1581]).

---

> ### Comment · Reviewer_8v9S · 2024-12-02
>
> > This paper does not propose new advanced machine learning methodology or demonstrate novel ML techniques.
>
> This is a silly, effortless feedback.
> Lot of progress in our field came from good benchmarks (examples: ImageNet, adversarial robustness evaluations circa '18, to more recent LLM evaluations such as chatbot arena, AlpacaEval and many others).
> If anything, I would say developing good benchmarks is at least, if not even more, important. For example, there were tons of papers claiming to tackle distribution shift pre-ChatGPT using various sophisticated new methodology, but comprehensive benchmarks like WILDS showed that most of the interventions don't generalize, and the best solution is to just finetune on OOD data (if you have them) most of the time.
>
> Also the work is clearly very well documented so not sure what the criticism about non-reproducibility is about..

---

### Official Review · Reviewer_zsfr · 2024-11-04

**Soundness:** 3
**Presentation:** 1
**Contribution:** 3
**Rating:** 6
**Confidence:** 4

**Summary:**

This work presents a benchmark for evaluating how well LLMs predict future events, curating questions about current events and corresponding answers. A set of state-of-the-art LLMs are then prompted to predict the probability of future events. The prediction accuracy is then compared against expert and non-expert human forecasters, ultimately finding that the LLMs generally perform similarly to average forecasters but worse than expert forecasters.

**Strengths:**

This paper has many strengths:
* Dynamic benchmarks are needed and event forecasting seems like a sensible LLM use to evaluate, likely interesting members of the research community
* This paper seems to describe a mature and already-used system that could elicit some interesting use
* The finding that LLMs can forecast events similarly to average forecasters is interesting, and the fact that experts are still better is a promising sign that this area deserves further attention
* The use of real human experiments is a big strength

**Weaknesses:**

This paper could be strengthened by addressing the following weaknesses:
1. The main weakness of this work is the limited experiment section. Given the main contribution is the curation of a new dataset and resultant evaluation, it is unsatisfying that the result is only one page of experiments with minimal interpretation. The result is it's unclear if this benchmark is useful beyond this experiment. Relatedly, it's never really shown why it's important for this to be a "dynamic" benchmark, despite this being a key selling point. Avoiding data leakage is indeed important, but it's surprising that this doesn't allow for any dedicated experiment or discussion around how often questions are resolved during the benchmarking.
2. The method development is described without many details, especially explanations of design decisions and their impact. Instead, most of Section 3 simply states what was done. It's not clear that future researchers could improve on this work if there's no explanation of what alternatives exist and why they are insufficient. This isn't inherently a major concern, but the main contribution of the work appears to be the dataset development, so defending these decisions would help the reader gain confidence in the resulting data and findings.
3. The paper is hard to follow, with many details sprinkled throughout various sections and the reader is left to hunt them down. It could help to include more details on the forecasting problem and summaries of exactly what's provided in Section 3.

**Questions:**

Answers to the following questions would help me better calibrate my review:
1. Is there intuition for why the LLMs would perform better on average for the questions sampled for humans? It seems like there's a consistent drop in the Brier Score.
2. How do LLM and human questions differ? Could these differences explain any differences in accuracy? Maybe this relates to L265---what's different about their information-availability?
3. What is "News with freeze values"?

---

> ### Author Response · Authors · 2024-11-24
>
> Thank you for your valuable feedback\! We address your questions and concerns below.
>
> ---
>
> ### Weaknesses
>
> **\> Weakness 1a. The main weakness of this work is the limited experiment section. Given the main contribution is the curation of a new dataset and resultant evaluation, it is unsatisfying that the result is only one page of experiments with minimal interpretation. The result is it's unclear if this benchmark is useful beyond this experiment.**
>
> A key topic of research is whether LLMs have achieved human parity in general reasoning, and if they have not, when or if they will. Forecasting is a useful testbed of LLM reasoning ability, just like math and coding. To produce an accurate forecast, a person or AI system must synthesize information, ideally from both live and preexisting knowledge, reason, avoid overconfidence, guard against biases, combine information, and quantify beliefs. Among the 17 models we test (across the 7 baselines outlined in Section 5.1), only Claude-3.5 Sonnet performed as well as the general public (Brier score of 0.111), and both Claude-3.5 and the general public perform significantly worse than superforecaster-level (Brier score of 0.091, lower is better). Our benchmark thus provides valuable insights into the capability of AI systems.
>
> We also address a key issue in LLM evaluation: the tendency for models to overfit to static datasets. For example, Zhang et al. https://arxiv.org/pdf/2405.00332 showed that on fresh GSM8K questions, model performance dropped by 13% for some families of models. This is no different in forecasting; a recent study claiming that LLMs are superhuman at forecasting (‘LLMs are Superhuman Forecasters’, by Phan et al.)) has been removed from arxiv after other researchers showed that the claims did not hold up when evaluated on a new set of questions.
>
> Here, since our benchmark is dynamic--automatically adding fresh forecasting questions about events that have not happened yet--we avoid potential leakage problems. We hope we set an example for the field to follow so we can more accurately assess models.
> Lastly, we recently analyzed the scaling trends of LLMs to observe how forecasting performance scales with more training compute. Figure 1 shows a negative log-linear relationship between training compute and forecasting accuracy (Brier score). This relationship was marginally statistically insignificant ($r = -0.64$, $p = 0.06$) because of the small number of tested models (only 9 models we tested had reliable estimates of training compute). But, we use this pattern to project out the log-linear relationship, finding that if performance continues to improve at these rates, LLMs could match superforecaster performance when training compute approaches $1 \times 10^{27}$ FLOP, though there is a large confidence interval (bootstrapped 95% CI: \[$7.08 \times 10^{26}$,$3.64 \times 10^{30}$\].
>
> We also analyzed the relationship between Chatbot Arena scores (see Chiang et al., 2024) and Brier scores, for which we could compare all 17 models tested. Here we found a strong and highly statistically significant relationship (r = -0.69, p = 0.002), showing that models with higher Arena scores also tend to produce more accurate forecasts. The calculated linear relationship implies that LLMs will match superforecaster performance when the Arena score approaches 1398 (bootstrapped 95% CI: [1335,1581]).
>
> We have added a paragraph entitled “LLM performance and forecasting accuracy” to Section 5.2 with this new analysis.

---

> ### Author Response · Authors · 2024-11-24
>
> **\> Weakness 1b. Relatedly, it's never really shown why it's important for this to be a "dynamic" benchmark, despite this being a key selling point. Avoiding data leakage is indeed important, but it's surprising that this doesn't allow for any dedicated experiment or discussion around how often questions are resolved during the benchmarking.**
>
> When we say dynamic, we mean a benchmark that is regularly updated so that there is a constant set of fresh, unresolved questions. Current forecasting datasets used to evaluate LLM forecasting capabilities rely on questions that have already been resolved. This has two major problems. First, they rely on the reported training cutoff of LLMs: they choose forecasting questions that resolve after training cutoff dates to evaluate forecasting ability; but these dates are not precise, and researchers cannot be sure these dates are accurate for closed-source models. Second, these datasets become outdated as groups release new models with training cutoffs after the resolution dates of the forecast questions.
>
> More generally, there's a well-established problem with data leakage in ML datasets. For example, Zhang et al. ([https://arxiv.org/pdf/2405.00332](https://arxiv.org/pdf/2405.00332)) showed that on fresh GSM8K questions, model performance dropped by 13% for some families of models. This is no different in forecasting; a recent study claiming LLMs are superhuman at forecasting real-world events (‘LLMs are Superhuman Forecasters’, by Phan et al.)) has been removed from arxiv after other researchers showed that the claims did not hold up when evaluated on a new set of questions.
>
> Since ForecastBench constantly adds questions about events that have not yet occurred, it is impossible for AI systems to ‘cheat.’ This is different from traditional ML datasets where the answers to the test set are either published or held by a third party. Our benchmark is valuable because it allows us to always be confident about the results of LLM forecasting capabilities. We hope we also set an example for the field to follow as it attempts to more accurately assess and compare models.
>
> In addition to the benefits above, we lastly highlight that a dynamic benchmark ensures that our question sets remain relevant to new events in the future, as the focus of world geopolitical, cultural, and economic events changes.
>
> \> **Weakness 1c. it's surprising that this doesn't allow for any dedicated experiment or discussion around how often questions are resolved during the benchmarking.**
>
> It is not clear to us what you mean by “how often questions are resolved during the benchmarking.” Data questions resolve at every forecast horizon once new data is available. Market questions resolve when the question is resolved on the platform they’re sourced from. Please let us know if this clarifies your understanding.

---

> > ### Author Response · Authors · 2024-11-24
> >
> > **\> Weakness 2\. The method development is described without many details, especially explanations of design decisions and their impact. Instead, most of Section 3 simply states what was done. It's not clear that future researchers could improve on this work if there's no explanation of what alternatives exist and why they are insufficient. This isn't inherently a major concern, but the main contribution of the work appears to be the dataset development, so defending these decisions would help the reader gain confidence in the resulting data and findings.**
> >
> > Thank you for this point. We have elaborated on these points in Section 3, explaining our methodological choices in more detail. We include the following information:
> >
> > - We rely on questions that cover a broad set of domains, but some domains are over-represented within the sources we sample from. So, when we construct the human question set from the LLM question superset, we first sample equally across sources and we then sample equally from domains within each source.
> > - We give humans 10 days to produce, update, and submit their forecasts because running surveys to get enough responses takes about that much time.
> > - We only provide LLMs with 24 hours to see the questions and forecast because we want to hamper potentially bad actors; if we were to give AI-based systems longer periods of time, dishonest actors could potentially supplement LLM forecasts with human forecasts. Importantly, LLMs and humans both submit their final forecasts on the exact same date with exactly the same availability of public information.
> > - For questions derived from datasets, we ask participants for forecasts across different time horizons so we can evaluate humans and LLM forecasting capabilities over the short-, medium-, and long-term.
> > - The human question set, contrary to the LLM set, does not contain combination questions. This is because the work of expert human forecasters is expensive and we could only rely on each expert forecaster submitting forecasts on 20-30 questions. We focused the experts on non-combination questions but plan to expand the human elicitation exercise to combination questions in future work.
> >
> > We hope these explanations in our updated draft describe the reasoning behind some of the decisions we made. If anything remains unclear, or if you think other points require further exposition, please let us know.
> >
> > **\> Weakness 3\. The paper is hard to follow, with many details sprinkled throughout various sections and the reader is left to hunt them down. It could help to include more details on the forecasting problem and summaries of exactly what's provided in Section 3\.**
> >
> > Thank you for the detailed feedback\! Given our page constraints and technical complexity, we strategically focused on key findings in the main text while placing implementation details in the appendix. For instance, we talk about surveying humans in section 4, but give elicitation details in Appendix D.
> >
> > We acknowledge the presentation could be enhanced and our revised draft addresses this concern.
> >
> > Concretely, we reorganized Section 3, removing unnecessary figures and breaking long blocks of text into paragraphs with headings while concentrating disparate bits of information in more logical places. This should make it easier to refer back to a given topic and find all pertinent information. We also modified the Baselines section slightly to have consistent names in the text and with the leaderboard table, which should make it easier to understand exactly how the baselines performed. We split the leaderboard table into two tables (Tables 2 and 3), one showing performance on the human question set of 200 questions and the other on performance of LLMs on the 1,000-item question set.
> >
> > We hope these changes address your concern.

---

> > > ### Author Response · Authors · 2024-11-24
> > >
> > > ### Questions
> > >
> > > **\> Question 1\. Is there intuition for why the LLMs would perform better on average for the questions sampled for humans? It seems like there's a consistent drop in the Brier Score.**
> > >
> > > The drop in Brier scores for LLMs on the LLM question set can primarily be attributed to the inclusion of combination questions, as these require reasoning about the interdependence of events—a task where current LLMs struggle. In contrast, the human question set contains only standard questions, which align more closely with typical forecasting tasks and the reasoning processes that LLMs are trained on. This alignment likely contributes to better LLM performance on these questions.
> > >
> > > Table 2 shows performance of each model and human comparison group on the 200 questions in the human question set, which only contains standard forecasting questions (no combination questions are in this set). Table 3 shows performance on the 1,000 questions of the LLM question set, which contains 500 combination questions.
> > >
> > > To compare LLM performance on the human subset of questions to LLM performance on the LLM subset of questions, please see Table 20 in Appendix H. This table shows performance on the 200 standard questions from the human question set plus the combination questions that exist in the LLM question set that are comprised of any 2 of those standard questions (Note: here the humans are at a disadvantage because we generate their forecasts for combination questions by treating their forecasts on the individual questions as independent). Now, comparing Table 20 to Table 3 shows that LLM performance doesn’t drop off when restricted to the human question set; rather, it drops off as a result of poor performance of LLMs on combination questions, which implies that forecasting the interaction of world events remains a challenging problem for LLMs.
> > >
> > > We updated Section 3 to make the question set composition more clear. We also moved Panel A and Panel B of Table 2 into their own tables (Table 2 and Table 3); we had originally grouped them together to stay within the page limit but, following some reorganization of the text, we decouple them for clarity.
> > >
> > > **\> Question 2\. How do LLM and human questions differ? Could these differences explain any differences in accuracy? Maybe this relates to L265---what's different about their information-availability?**
> > >
> > > As explained above, the human question set of 200 questions excludes combination questions whereas the LLM question set of 1,000 questions contains 500 combination questions. We exclude combination questions from the human question set because:
> > >
> > > - Expert human forecasts are expensive to generate, and by focusing on forecasts of the main indicators we maximize the relevance of these expert human forecasts.
> > > - We can always interpolate human performance on combination questions based on their forecast of individual questions: assume every pair of questions is independent, and simply multiply the probabilities to arrive at a forecast of the combination question. While this forecast is less than ideal, we can use it to provide a lower bound of human accuracy. On the other hand, if we had elicited forecasts of combination questions, we could not easily back out expert human forecasts of the underlying indicators.
> > >
> > > In future work, we plan to survey humans directly on the combination questions themselves.
> > >
> > > **\> Question 3\. What is "News with freeze values"?**
> > >
> > > There is a missing baseline in Section 5.1. Thanks for catching this. In the updated draft, we changed baseline 5 from “scratchpad with information retrieval” to “scratchpad with news,” to better correspond to the terminology in Tables 2 and 3\. We also added a 6th baseline: “scratchpad with news with freeze values,” that supplements the scratchpad prompt with both news articles and the market values on the question set freeze date.
> > >
> > > ---
> > >
> > > Please let us know if you have any other questions or concerns we can address. If not, we hope that you will consider increasing your rating.

---

> > > > ### Comment · Reviewer_zsfr · 2024-11-26
> > > >
> > > > Thanks for your efforts in addressing my concerns. The paper is certainly improved, especially with better clarity in the revised paper. I remain surprised that despite this being a dynamic benchmark, there are not experiments showing any results that change over time and that there are not demonstrations that non-dynamic benchmarks fail, empirically motivating the need to avoid leakage (which I agree is needed). Nevertheless, I believe enough of my concerns are resolved so I have raised my score.

---

### Official Review · Reviewer_4xz8 · 2024-11-04

**Soundness:** 3
**Presentation:** 3
**Contribution:** 3
**Rating:** 8
**Confidence:** 4

**Summary:**

This paper proposes a forecasting benchmark for evaluating LLM's forecasting capability. The benchmark is carefully constructed to avoid potential data leakage. The benchmark results indicates with statistical significance that even the SOTA LLMs nowadays cannot beat human experts in this task.

**Strengths:**

1. The forecasting problem studied in this paper is both very interesting and practical. Forecasts on macroeconomic problems such as presidential election is both complicated and impactful to real-world scenarios. Hence it can serve as a good benchmark for evaluating LLM's capability.

2. The benchmark design considers data leakage, which improves the reliability of the benchmark.

3. The benchmark results indicates some areas where LLMs still have inferior performance compared to top human, revealing potential future research directions.

4. Code is provided for better reproducibility.

**Weaknesses:**

1. It may be worth justifying the adopted method in this benchmark for LLM to make forecasts, since LLM reasoning methods (e.g. RAG, etc.) can have great influence on the final performance

**Questions:**

1. What if we allow LLMs to access different tools to actively acquire relevant information in a feedback loop? In this way the LLM may construct more context for more accurate answer.

---

> ### Author Response · Authors · 2024-11-24
>
> Thank you for reviewing and engaging with our work\! We address your questions and concerns below.
>
> ---
>
> ### Weaknesses
>
> **\> It may be worth justifying the adopted method in this benchmark for LLM to make forecasts, since LLM reasoning methods (e.g. RAG, etc.) can have great influence on the final performance**
>
> The choice of methods and prompts indeed significantly affects LLM forecasting performance.
>
> For our initial evaluation, we adopted the methodology from Halawi et al. (2024), a recent paper that established effective practices for LLM forecasting, including both prompting strategies and question filtering criteria.
>
> While this approach helped demonstrate how different reasoning methods and model choices influence forecasting accuracy, its primary value lies in establishing a consistent baseline for measuring improvements in LLM forecasting capabilities over time; maintaining the same methods and varying only the model used will allow us to do this.
>
> That said, we’re excited to see teams experiment with their own approaches by submitting forecasts on ForecastBench.
>
> ### Questions
>
> **\> What if we allow LLMs to access different tools to actively acquire relevant information in a feedback loop? In this way the LLM may construct more context for more accurate answer.**
>
> This approach, which aligns with the agentic LLM paradigm, is indeed a promising direction. A competitive forecasting system would likely benefit from accessing relevant tools, reasoning iteratively, updating its memory, and refining outputs through a feedback loop. We anticipate improved performance with this paradigm for two reasons: (1) it closely mirrors human processes for forecasting, and (2) it has demonstrated superior results on various benchmarks, including SWE-Bench and Web-Bench (Wang et al. [https://arxiv.org/pdf/2407.16741](https://arxiv.org/pdf/2407.16741)), CYBench (Zhang et al. [https://arxiv.org/pdf/2408.08926](https://arxiv.org/pdf/2408.08926)), and others.
>
> While we recognize the potential of this approach, developing such a system would require substantial effort and resources. For the scope of our current work, we believe our existing baselines, particularly Halawi et al. (2024), provide a strong foundation for comparison. We have thus left the exploration of this agentic paradigm to future research directions.
>
> ---
>
> Please let us know if you have any other questions or concerns we can address.

---

### Official Review · Reviewer_UK3t · 2024-11-04

**Soundness:** 3
**Presentation:** 4
**Contribution:** 4
**Rating:** 8
**Confidence:** 4

**Summary:**

The paper introduces ForecastBench, a novel dynamic benchmark designed to evaluate the forecasting capabilities of AI systems, particularly large language models. In contrast to prior static benchmarks, ForecastBench continuously updates with new forecasting questions about future events that have no known answers at the time of submission, thereby eliminating concerns of data leakage or contamination. The study compiles a standardized set of 1000 forecasting questions sourced daily from nine data providers, including prediction markets, forecasting platforms, and real-world time series. To evaluate the accuracy, the authors assess the performance of 17 state-of-the-art LLMs on a subset of these questions, comparing their forecasts to those of expert human forecasters (superforecasters) and the general public, using the Brier score as the primary metric.

**Strengths:**

- A sophisticated evaluation framework that incorporates multiple forecasting dimensions, including:
    - Selection of questions from high-liquidity market sources,
    - Temporal sampling methodology for question release,
    - Granular analysis of model and human performance across varying time horizons.
    - Comprehensive assessment of instruction-following chat models through six distinct evaluation methodologies (Section 5.1), including zero-shot prompting, scratchpad reasoning prompting, and news-augmented generation.
- Implementation of diverse temporal resolutions for forecasting questions, spanning from week-long to decade-scale predictions, enabling robust evaluation across different time scales.
- Novel incorporation of combination questions for assessing model performance on higher-complexity tasks, revealing significant human performance advantages and highlighting current limitations in model capabilities.
- The development of a daily question-release mechanism featuring 1000 forecasting queries, which:
    - Mitigates potential gaming of the leaderboard system,
    - Generates an expanding dataset of resolved predictions enabling model training and calibration of future forecasting systems.

**Weaknesses:**

- The evaluation of unresolved questions relies on comparisons with crowd predictions, potentially complicating future assessments of whether models outperform the crowd baseline.
- The number of expert predictions was limited, with only 39 experts contributing, leading to an average of 8 expert predictions per question. This relatively small sample size may affect the robustness and generalizability of the evaluation results.

**Questions:**

- Have you investigated the impact of the prompt date on model predictions? Specifically, could questions asked further in advance of the resolution date introduce greater uncertainty in LLM predictions, potentially affecting prediction accuracy?
- How is the quality of questions released for evaluation determined?
- Have you considered evaluating model predictions by examining the logits of the answer tokens? Although this may be more feasible with open-source models, some providers do offer token-level logit outputs. Incorporating logits could provide insights into model calibration and further enhance prediction reliability.

**Details Of Ethics Concerns:**

- Have you investigated the effect of the prompt date on model predictions? Specifically, could questions asked further in advance of the resolution date introduce greater uncertainty into LLM predictions, potentially affecting prediction accuracy?
- How is the quality of questions released for evaluation determined?
- Have you considered evaluating model predictions by examining the logits of the response tokens? Although this may be more feasible with open-source models, some providers offer token-level logit output. Including logits could provide insight into model calibration and further improve prediction reliability.

---

> ### Author Response · Authors · 2024-11-24
>
> Thank you for reviewing and engaging with our work\! We address your questions and concerns below.
>
> ---
>
> ### Weaknesses
>
> **\> The evaluation of unresolved questions relies on comparisons with crowd predictions, potentially complicating future assessments of whether models outperform the crowd baseline.**
>
> In many markets, the crowd forecast approaches the eventual market resolution as the resolution date approaches. Hence, evaluating unresolved market questions against the crowd forecast is the best estimate we can obtain of their accuracy before the question is resolved.
>
> Once a question has been resolved, the forecast is compared to ground truth. We can always compare model performance to the crowd forecast at any point in time if we so desire. Indeed, in every forecasting round we run a dummy forecasting baseline called the “imputed forecaster,” that forecasts the crowd forecast on the forecast due date and 0.5 on dataset questions
>
> As a corollary, on the ForecastBench website we allow viewers to sort the leaderboard by performance only on resolved questions.
>
> **\> The number of expert predictions was limited, with only 39 experts contributing, leading to an average of 8 expert predictions per question. This relatively small sample size may affect the robustness and generalizability of the evaluation results.**
>
> We agree that the number of expert predictions was limited by our small sample of experts. But we also believe that this provides a realistic measure of the level of state-of-the-art human forecasting ability. A small number of firms and think tanks provide expert forecasting in geopolitical and economic domains where human forecasters with proven track records answer policy-relevant questions. It is not possible or cost-effective to get 10+ forecasters to submit forecasts on even a handful of questions. In the private market, doing this for even one question would cost thousands of dollars. So, our goal with the expert human forecaster comparison group was to provide a realistic measure of accuracy from the current top-level human forecasters who might be asked to answer questions like these. We felt that getting around 5-10 of these forecasters per question accomplished our goal there.
>
> ### Questions
>
> **\> Have you investigated the impact of the prompt date on model predictions? Specifically, could questions asked further in advance of the resolution date introduce greater uncertainty in LLM predictions, potentially affecting prediction accuracy?**
>
> We expect this to be the case and plan to evaluate long-term forecasting accuracy of both LLMs and humans in follow-up studies. This is one reason the questions sourced from datasets are important: for these questions, we ask for forecasts at 8 intervals, from 7 days to 10 years into the future. Over time, we’ll be able to assess and compare forecasting ability across time horizons amongst benchmark forecasters and LLM baselines. We can also explore whether alternative prompting strategies reduce any obvious patterns of bias in the elicitation of forecasts over different time horizons relative to the prompt date.
>
> We have clarified this point in Section 3.2 in the Forecast horizons paragraph.
>
> **\> How is the quality of questions released for evaluation determined?**
>
> We follow the method outlined in Halawi et al. (2024) to filter out low-quality questions, prompting gpt-3.5-turbo-0125 with a question validation prompt that includes a quality rubric. We have added the question validation prompt we use to Appendix F figure 10\.
>
> **\> Have you considered evaluating model predictions by examining the logits of the answer tokens? Although this may be more feasible with open-source models, some providers do offer token-level logit outputs. Incorporating logits could provide insights into model calibration and further enhance prediction reliability.**
>
> We have not done this yet but agree that this would be a worthwhile addition to a follow-up paper.
>
> ---
>
> Please let us know if you have any other questions or concerns we can address.

---

> > ### Comment · Reviewer_UK3t · 2024-11-25
> >
> > Thank you for addressing the questions and updating the paper. I have no further comments.

---

### Official Review · Reviewer_8v9S · 2024-11-08

**Soundness:** 3
**Presentation:** 4
**Contribution:** 3
**Rating:** 8
**Confidence:** 4

**Summary:**

The paper presents a sophisticated and automated benchmark for evaluating forecasts that improves over prior benchmarks in that it will not become stale and does not rely on assumptions about models (e.g., knowledge cutoffs). Using the benchmark, they find that interestingly, SOTA models are still short of expert forecasters.

**Strengths:**

- very well written--well motivated, structured, and polished overall
- addresses a key problem in prior benchmarks: benchmark staleness, which is obviously very important for the problem of forecasting, since the goal is to build/evaluate systems that can predict new events, not post-hoc. their approach is conceptually cleaner (imo) than relying on LLMs' knowledge-cutoffs, etc.
- overall interesting finding that at least in this setting, LLMs still quite lag behind experts, only matching the median of average people
- the approach to building/generating the question set and the entire pipeline is very thoroughly thought out.

**Weaknesses:**

- Table 2 is a bit big / dense and hard to parse. The paper could benefit from using more visual plots to convey the results

**Questions:**

- Is there anything easy (cheap) we can try to improve the capabilities of these models to forecasting beyond RAG-like things? I guess finetuning open-source models would be ideal but that is probably infeasible.

---

> ### Author Response · Authors · 2024-11-24
>
> Thank you for reviewing and engaging with our work\! We address your questions and concerns below.
>
> ---
>
> ### Weaknesses
>
> **\> Table 2 is a bit big / dense and hard to parse. The paper could benefit from using more visual plots to convey the results**
>
> We agree that it’s quite dense. In our updated version of the paper, we streamlined the text and were able to split up the table (now Table 2 and Table 3) while staying within the page limit. We have also added plots (Figures 1a and 1b) showing the relationship between performance on ForecastBench and training compute/Chatbot Arena scores.
>
> ### Questions
>
> **\> Is there anything easy (cheap) we can try to improve the capabilities of these models to forecasting beyond RAG-like things? I guess finetuning open-source models would be ideal but that is probably infeasible.**
>
> Yes. Recent work by Halawi et al. (2024) finetuned a reasoning model (from GPT-4) precisely to reason and make judgmental forecasts. Their result shows that even with finetuning and RAG, LLMs still did not surpass or reach human performance, but using such methods helped improve performance quite a lot (decreased Brier score by 0.03 with respect to the baseline).
>
> However, our work aims to establish a benchmark and a set of baselines. We believe it is possible to improve upon our baselines via finetuning using more advanced LLMs other than GPT-4, but we will leave this to future work.
>
> ---
>
> Please let us know if you have any other questions or concerns we can address.

---

> > ### Comment · Reviewer_8v9S · 2024-12-02
> >
> > Thank you for the answers and edits to the paper.
> > I think it might be worth trying to bring up Figures 1a/b to earlier in the paper. It is a really nice figure, and also the paper is a bit lacking in visual until then.
> >
> > I remain largely very positive of the paper!

---

### Official Review · Reviewer_tDYK · 2024-11-08

**Soundness:** 2
**Presentation:** 3
**Contribution:** 2
**Rating:** 5
**Confidence:** 4

**Summary:**

A dynamic dataset of forecasting qs about the future is aggregated and proposed as a benchmark for LLMs. Performance of a variety of LLMs is compared with human.

**Strengths:**

Interesting and useful benchmark, whose position in the landscape is clearly described.
LLM evals are extensive.

**Weaknesses:**

This is an interesting and useful dataset, but reads like a tech report or class assignment; the level of technical detail is superficial, and there is little to no exploration of what we learn about LLMs from this.

**Questions:**

The level of technical details and insight are the biggest things that would make me change my score.

Detailed comments by section:

abs/intro/relwork
 - "necessary task" seems a bit overstated; if you want to make this claim it would be worth stating how exactly the referenced papers support that claim
- reasonably clear intro/positioning and motivation overall; a bit repetitive but lacking technical details (e.g. how are predictions elicited from humans? what kinds of mistakes do LLMs make?); it would be nice to mention that briefly
- relwork: what is "statistically imprecise"?  - good lmeval section
preliminaries
 - unclear details about the nature of forecasts in the dataset; are they open-ended questions / natural language? are the possible outcomes enumerated in advance? is there a char limit? etc.  - unclear explanation of the Brier score; the Forecasting paragraph says forecasters "may" assign probabilities to possible outcomes; if they don't how do you assign f? How do you assign a 0/1 outcome to questions like "who will be the next US president?" - using the character f is confusing notation (f is almost universally a function) but not a huge deal - "strictly proper" and why we would care about that / why that incentivises truthfulness is not explained at all.  - 3 different methods of prompting/rag are not explained at all - "prompting" for the humans that are compared to the LLMS is not explained at all

sec 3
 - does not seem necessary to have a flowchart here to say that everything goes to the question bank, and it takes a lot of space. It might be helpful to have one if it showed what are the different platforms and datasets, any pre-preprocessing that needs to be done, etc., but this is covered fine in Table 1.
 - Similarly for Fig 3; neither of these is contributing substantively to the paper; they could be moved to appendix to make space for more technical details.
 - it seems like the details in 3.2 contradict the "preliminaries" section

sec 5
- given the small dataset sizes here it would be interesting to explore exactly what LLMs do and don't get right

---

> ### Author Response · Authors · 2024-11-24
>
> Thank you for your valuable feedback\! We address your questions and concerns below.
>
> ---
>
> ### Weaknesses
>
> **\> This is an interesting and useful dataset, but reads like a tech report or class assignment; the level of technical detail is superficial, and there is little to no exploration of what we learn about LLMs from this.**
>
> A key topic of research is whether LLMs have achieved human parity in general reasoning, and if they have not, when or if they will. Forecasting is a useful testbed of LLM reasoning ability, just like math and coding. To produce an accurate forecast, a person or AI system must synthesize information, ideally from both live and preexisting knowledge, reason, avoid overconfidence, guard against biases, combine information, and quantify beliefs. Among the 17 models we test (across the 7 baselines outlined in Section 5.1), only Claude-3.5 Sonnet performed as well as the general public (Brier score of 0.111), and both Claude-3.5 and the general public perform significantly worse than superforecaster-level (Brier score of 0.091, lower is better). Our benchmark thus provides valuable insights into the capability of AI systems.
>
> We also address a key issue in LLM evaluation: the tendency for models to overfit to static datasets. For example, Zhang et al. [https://arxiv.org/pdf/2405.00332](https://arxiv.org/pdf/2405.00332) showed that on fresh GSM8K questions, model performance dropped by 13% for some families of models. This is no different in forecasting; a recent study claiming that LLMs are superhuman at forecasting (‘LLMs are Superhuman Forecasters’, by Phan et al.)) has been removed from arxiv after other researchers showed that the claims did not hold up when evaluated on a new set of questions.
>
> Here, since our benchmark is dynamic \-- automatically adding fresh forecasting questions about events that have not happened yet \-- we avoid potential leakage problems. We hope we set an example for the field to follow so we can more accurately assess models.
>
> Lastly, we recently analyzed the scaling trends of LLMs to observe how forecasting performance scales with more training compute. Figure 1b shows a negative log-linear relationship between training compute and forecasting accuracy (Brier score). This relationship was marginally statistically insignificant ($r=-0.64$, $p=0.06$) because of the small number of tested models (only 9 models we tested had reliable estimates of training compute). But, we use this pattern to project out the log-linear relationship, finding that if performance continues to improve at these rates, LLMs could match superforecaster performance when training compute approaches $1 \\times 10^{27}$ FLOP, though there is a large confidence interval (bootstrapped 95% CI: \[$7.08 \\times 10^{26}$,$3.64 \\times 10^{30}$\]. As we continue to add more models, we'll be able to refine our scaling trends.
>
> We also analyzed the relationship between Chatbot Arena scores (see Chiang et al., 2024\) and Brier scores, for which we could compare all 17 models tested. Here we found a strong and highly statistically significant relationship ($r=-0.69$, $p=0.002$), showing that models with higher Arena scores also tend to produce more accurate forecasts. The calculated linear relationship implies that LLMs will match superforecaster performance when the Arena score approaches 1398 (bootstrapped 95% CI: \[1335,1581\]).
>
> We have added a paragraph entitled “LLM performance and forecasting accuracy” to Section 5.2 with this new analysis.

---

> > ### Author Response · Authors · 2024-11-24
> >
> > ### Questions
> >
> > **\> abs/intro/relwork  \- "necessary task" seems a bit overstated; if you want to make this claim it would be worth stating how exactly the referenced papers support that claim**
> >
> > Forecasting is often necessary in considering the consequences of a set of actions under consideration. In the paper we gave two intuitive instances of this: hiring decisions and the COVID response.
> > * Christensen et al. (2018) demonstrate how economic forecasts influence investment and hiring decisions.
> > * Adam (2020) demonstrates that forecasts of the Covid-19 pandemic in early 2020 are what prompted local lockdowns to slow the spread of the virus.
> >
> > There are many other situations where organizations act based on forecasts: international aid agencies forecast extreme weather events (droughts, storms, etc.), taking anticipatory action to mobilize relief efforts (e.g. https://www.unocha.org/anticipatory-action); venture capitalists forecast tangible (e.g. market conditions) and intangible (e.g. effectiveness of leadership of a company) things to decide whether or not to invest in a company; and Barcelona invested in a 13-year old Lionel Messi because they believed he would, one-day, be good enough to play on the first team, and so on. Though some forecasts are made formally by assigning a probability to the likelihood of an outcome of key decisions, others are made using qualitative probabilistic expressions (such as “likely to occur,” “probably won’t happen,” etc.). These are all still forecasts that guide decisions.
> >
> > We agree with your point that "necessary task" might be too strong, since perhaps there are exceptions. We have changed the wording to “important task” and appreciate the feedback.
> >
> > **\> reasonably clear intro/positioning and motivation overall; a bit repetitive but lacking technical details (e.g. how are predictions elicited from humans? what kinds of mistakes do LLMs make?); it would be nice to mention that briefly**
> >
> > We survey the general public on Qualtrics and superforecasters on Quorum. Screenshots taken from the survey software of sample questions are in Figures 2 and 3\. For each question, humans are prompted to provide a forecast in \[0,100\] representing the probability of an event occurring. They’re also invited to provide the rationale underlying their forecast.
> >
> > These details along with further information on how forecasts were elicited can be found in Section 4 and Appendix D. Please let us know if there is any additional information you are looking for.
> >
> > Regarding mistakes that LLMs make, please see the response to the last question below for more on this.
> >
> > **\> relwork: what is "statistically imprecise"?**
> >
> > In the context of the sentence, “statistically imprecise” means that because there were very few (only 31\) forecasting questions in the tournament studied by Schoenegger et al. (2024b), the authors can't be confident of their main result: that an ensemble of LLMs was as accurate as human forecasters. While the comparison showed no clear difference between the performance of LLMs and humans, it remained quite underpowered and could not have detected meaningful differences in performance. That’s one reason why our benchmark, with thousands of forecasts, enables more definitive conclusions about LLM forecasting capabilities.
> >
> > We have further clarified this in the updated paper.

---

> > > ### Author Response · Authors · 2024-11-24
> > >
> > > **\> \- good lmeval section preliminaries \- unclear details about the nature of forecasts in the dataset; are they open-ended questions / natural language? are the possible outcomes enumerated in advance? is there a char limit? etc.**
> > >
> > > Regarding forecast questions, examples of questions that were included in the survey are:
> > > 1. Will the Cavendish account for less than 50% of banana exports worldwide before 2035?
> > > 2. What is the probability that the daily average temperature at the French weather station at Abbeville will be higher on \`{resolution\_date}\` than on \`{forecast\_due\_date}\`?
> > > 3. Will a politician claim they lost a major election due to a “deepfake” image, video, or audio recording in a G20 country before 2025?
> > > 4. "According to Wikipedia, will Sarasadat Khademalsharieh have an Elo rating on \`{resolution\_date}\` that's at least 1% higher than on \`{forecast\_due\_date}\`?"
> > >
> > > These are presented in Tables 8 and 9, and Figures 2 and 3\. Tables 8 and 9 show the question data as it’s provided in the question set. Figures 2 and 3 are screenshots of this data as displayed to humans in the survey.
> > >
> > > The possible outcomes for each question upon event resolution are either 0 or 1, where 0 is No and 1 is Yes. From Section 2 (Metrics paragraph), forecasts are real numbers in \[0,1\]. Forecasts are accepted in JSON format so they’re passed as JSON numbers. LLMs are prompted to produce a forecast in \[0,1\]. See, for example, this text from the zero-shot prompt:
> > >
> > > > Make a prediction of the probability that the question will be resolved as true. You MUST give a probability estimate between 0 and 1 UNDER ALL CIRCUMSTANCES. If for some reason you can't answer, pick the base rate, but return a number between 0 and 1.
> > >
> > > For the full text of all prompts used, see Appendix F.
> > >
> > > There’s no char limit for the fields of the question set file. The only time we ran into context window limits is when we supplemented questions with news stories. The questions with lengths of higher variance are the market questions as the dataset questions follow a template. Below is a list of average question length and maximum question length for all question sources. Here, question length refers to the length of all text contained as a value from all fields in the data dictionary shown in Table 5 of Appendix B.1.
> > >
> > > | Source | Average Length | Max Length |
> > > |--------|---------------|------------|
> > > | manifold | 1,230 | 6,539 |
> > > | metaculus | 2,767 | 6,176 |
> > > | infer | 3,330 | 7,749 |
> > > | polymarket | 1,387 | 1,764 |
> > > | acled | 2,567 | 2,658 |
> > > | dbnomics | 1,194 | 1,247 |
> > > | fred | 2,182 | 7,055 |
> > > | wikipedia | 1,898 | 2,633 |
> > > | yfinance | 2,387 | 3,016 |
> > > | **Overall Average** | **2,104** | **7,749** |
> > >
> > > **\> \- unclear explanation of the Brier score; the Forecasting paragraph says forecasters "may" assign probabilities to possible outcomes; if they don't how do you assign f?**
> > >
> > > There are many possible ways to score forecasts, but perhaps the most common and widely-used is the Brier score. It’s easiest to understand the Brier score as the mean squared error applied to forecasts, with penalties increasing for a forecast being farther away from the final outcome.
> > >
> > > Regarding missing forecasts, we indeed account for those as LLMs may not return an answer, and discuss this in Section 3.3. For missing forecasts on dataset questions, we fill in the value 0.5, which is equivalent to saying you don’t know whether the question will resolve positively or negatively. For missing forecasts on market questions, we fill in the market value on the day forecasts were due; this is because a dishonest forecasting team, instead of using LLMs, could just get the latest crowd forecast from the market and use that and we didn’t want teams to perform well by doing this.
> > >
> > > If anything is unclear or you disagree with how we handle missing forecasts, we are happy to address your concerns.
> > >
> > > **\> How do you assign a 0/1 outcome to questions like "who will be the next US president?"**
> > >
> > > We agree this is unclear and have replaced the example with “Will a Democrat win the 2028 US presidential election?” in the updated draft. Thank you for catching this.
> > >
> > > **\> \- using the character f is confusing notation (f is almost universally a function) but not a huge deal**
> > >
> > > We appreciate your point and agree that $f$ is almost universally a function. Though we'd like to note that $f$ is also used to represent a forecast both on the Brier score wiki page and in Brier’s original paper introducing the scoring rule, so we stick with this convention.

---

> > > > ### Author Response · Authors · 2024-11-24
> > > >
> > > > **\> \- "strictly proper" and why we would care about that / why that incentivises truthfulness is not explained at all.**
> > > >
> > > > The Brier score is strictly proper because the value is minimized in expectation when the forecasted value is equal to the forecaster's true belief about the probability. As the forecaster wants to obtain the lowest (best) Brier score to be ranked highest on the leaderboard, they have no incentive to misreport their beliefs; being overconfident or underconfident would only worsen their expected score.
> > > >
> > > > As an example of an improper scoring rule, let’s say if you bet correctly on the desired outcome you will gain 1 point, and if you were wrong you will lose 1 point.
> > > >
> > > > Suppose for every single event with an actual probability of 60%, you are perfectly calibrated and know this, but Jean always thinks the probability is 100%. Under our naive draft, you two will always bet the same way, “yes”, in these events, but the relative skill between you will never be discovered.
> > > >
> > > > We want a strictly proper scoring rule; that is, if you think the reality of the world is $q$ then your prediction should actually be equal to $q$ to maximize your expected score.
> > > >
> > > > The Brier score incentivizes honest reporting of probabilities, and this is what a strictly proper scoring rule means.
> > > >
> > > > **\> \- 3 different methods of prompting/rag are not explained at all**
> > > >
> > > > The three methods of prompting/RAG we used are:
> > > >
> > > > 1. _Zero-shot prompting_: In this approach, the model generates a forecast directly without producing intermediate content, such as step-by-step reasoning. The prompt for this method is provided in Figure 4 in Appendix F.
> > > > 2. _Scratchpad instructions_: These prompts, based on the method described by Nye et al. (2021), guide the model through a structured reasoning process. Our implementation follows Halawi et al. (2024), where prompts were iteratively refined by analyzing the Brier score after each modification and addressing common LLM errors, such as misinterpreting questions. The prompt used for this method is included in Figure 5 in Appendix F.
> > > > 3. _Scratchpad instructions with information retrieval_: This method builds on the second approach by incorporating relevant news articles to provide additional context to the model. The scratchpad prompt remains the same as in method 2, supplemented with retrieved information.
> > > >
> > > > We ran the three baselines above both with and without incorporating “freeze values.” For market questions, freeze values represent the crowd forecasts available on the market on the day the question set was created. For dataset questions, these represent a reference value useful in the forecasting task.
> > > >
> > > > Additionally, we ran a simulated LLM “crowd” forecast as another baseline. This method used three different superforecaster scratchpad prompts (Figures 6, 7, and 8 in Appendix F) across three models to generate forecasts. The outputs were aggregated to form a single LLM “crowd” forecast baseline.
> > > >
> > > > These methods are detailed in Section 5.1 under the Baselines paragraph, which outlines seven specific points. Further information about the LLM crowd forecast can be found in Appendix E, and the exact prompts are provided in Appendix F. We have clarified the Baselines paragraph in the updated draft, but if anything remains unclear or requires additional elaboration, please let us know.
> > > >
> > > > **\> \- "prompting" for the humans that are compared to the LLMS is not explained at all**
> > > >
> > > > From Appendix D, human forecasters are prompted as:
> > > >
> > > > > You are going to be predicting the probability of the answer to the question below being "Yes" (or "resolving positively").
> > > >
> > > > For examples of questions provided to humans and the complete instructions they received, see Appendix D. Specifically, Figures 2 and 3 are screenshots of example questions from the survey. We include these details in the appendix due to limited space in the body of the paper and explaining these nuances in the main body would make the paper hard to read.
> > > >
> > > > **\> sec 3   \- does not seem necessary to have a flowchart here to say that everything goes to the question bank, and it takes a lot of space. It might be helpful to have one if it showed what are the different platforms and datasets, any pre-preprocessing that needs to be done, etc., but this is covered fine in Table 1\.**
> > > >
> > > > We agree with you and appreciate the feedback. We removed this and other figures that were described in the text.

---

> > > > > ### Author Response · Authors · 2024-11-24
> > > > >
> > > > > **\> Similarly for Fig 3; neither of these is contributing substantively to the paper; they could be moved to appendix to make space for more technical details.  \- it seems like the details in 3.2 contradict the "preliminaries" section**
> > > > >
> > > > > We agree with you and appreciate the feedback. We removed this and other figures that were described in the text.
> > > > >
> > > > > Do you say that the details in Section 3.2 contradict the “preliminaries” section because we evaluate unresolved forecast questions by using the crowd forecast whereas we evaluate resolved forecast questions using ground truth (via the Brier score)? If so, we don’t view this as a contradiction. In many instances, crowd forecasts on active markets tend toward ground truth as the resolution date approaches, so evaluating against the crowd forecast can provide a good estimate of forecast performance while we wait for the questions to resolve. This allows us to have a signal about forecasting performance on market questions before question resolution. That said, all forecasts on market questions are eventually resolved to ground truth.
> > > > >
> > > > > We have tried to clarify this distinction in the updated draft.
> > > > >
> > > > > If we are misunderstanding and you find something else to be contradictory, we can certainly address it.
> > > > >
> > > > > **\> given the small dataset sizes here it would be interesting to explore exactly what LLMs do and don't get right**
> > > > >
> > > > > Thanks for the suggestion. To investigate further, we broke down performance by question category, Table 25, and forecast horizon, Table 26\. We do this for the best-performing LLM, Claude Sonnet 3.5 (scratchpad prompt with freeze values).
> > > > >
> > > > > Superforecasters outperform Claude Sonnet 3.5 on all question categories except Environment & Energy and Sports. Interestingly, superforecasters outperform Claude Sonnet 3.5 at all forecast horizons.
> > > > >
> > > > > We added Appendix M to the paper to highlight these results.
> > > > >
> > > > > ---
> > > > >
> > > > > Please let us know if you have any other questions or concerns we can address. If not, we hope that you will consider increasing your rating.

---

### Author Response · Authors · 2024-11-26

We thank the reviewers for their thoughtful comments and constructive feedback. We have responded to reviewer questions and made corresponding changes to our submission. We discuss the goals of this project and our key edits to the paper, below.

As with math and coding, forecasting provides a useful testbed for LLM reasoning capabilities. Designed as a dynamic benchmark with regularly and automatically updated question sets, resilient to data leakage and benchmark staleness (as noted by Reviewers 8v9S, 4xz8, and zsfr), we believe that ForecastBench is well-suited to serve as the reference benchmark for LLM forecasting abilities.

We’re excited that ForecastBench provides a sophisticated and well-thought-out pipeline with which to evaluate LLM forecasting capabilities, as noted by Reviewers 8v9S, UK3t, and zsfr.

Our results allow for the comparison of LLMs to both the general public and superforecasters, showing that LLMs lag behind expert forecaster performance. This benchmark therefore identifies a potential avenue for LLM improvement in a complex reasoning task, as noted by Reviewers 4xz8, 8v9S, UK3t, and zsfr.

We want to highlight the following changes made in response to reviewer comments. **We mark significant changes to the revised PDF in yellow for ease of reference.**

* **Expanded Results section**: We significantly expanded our Results and Discussion sections (sections 5.2 and 6) to highlight how this work deepens our understanding of LLMs.
* **Explored scaling trends**: We include new analyses on scaling trends and on the relationship between model performance on forecasting questions and established benchmarks/training compute in Section 5.2.
* **LLM performance breakdown**: We provide a detailed breakdown of LLM and human performance across question categories and forecast horizons (Appendix M). This analysis highlights areas where LLMs excel or struggle.
* **Demographic data**: We added detailed demographic information about the human survey participants to provide a clear context for the comparisons between human and LLM forecasts (Appendix L).
* **Updated tables and results**: We updated tables and results to reflect the latest leaderboard standings, now evaluated at 7-, 30-, and 90-day forecast horizons. These updates ensure the paper is fully aligned with the most recent information on forecasting ability.
* **Expository improvements**: We clarified several aspects of our writing, including:
  * The differences between human and LLM question sets.
  * Our methodology for resolving unresolved market questions.
  * Removal of redundant information and figures.
  * Reorganization of the paper to improve structure and readability.
  * Splitting the leaderboard results into separate tables (now Tables 2 and 3) to provide clearer perspectives of performance on human and LLM question sets.

We have responded to the initial comments of each reviewer below and kindly encourage reviewers to reconsider their ratings if they believe their concerns have been adequately addressed.

**We hope our answers resolve all initial questions and concerns raised by the reviewers and we are more than happy to answer any remaining questions during the discussion phase\!**

---

### Meta-Review · Area_Chair_FrZp · 2024-12-20

**Metareview:**

This well-written and well-motivated paper proposes a new benchmark for LLM-based forecasting that aims to mitigate the staleness of existing benchmarks. It has been evaluated by 6 knowledgeable reviewers. Most of them recommended its acceptance (including 2 straight acceptances and 1 marginal acceptance), while other 2 voted for marginal rejection. The authors have provided an extensive rebuttal and substantially improved the paper in response to the reviews. Based on the maturity of the presented work and its scope, I recommend accepting this work for publication at ICLR.

**Additional Comments On Reviewer Discussion:**

Some reviewers, perhaps overly, hoped to see novel ML methodology, but this paper developed a new benchmark. Most of the reviewers have recognized that and focused on that in their assessment. Robust benchmarks are sorely needed to demonstrate success and to guide further development of AI.

---

### Decision · Program_Chairs · 2025-01-22

Accept (Poster)